# SEOUL AR: Designing a Mobile AR Tour Application for Seoul Sky Observatory in South Korea

**Soomin Shin and Yongsoon Choi ***

Department of Art & Technology, Sogang University, Seoul 04107, Korea; soomin011@gmail.com
* Correspondence: yongsoon@sogang.ac.kr; Tel.: +82-2-3274-4893

**Abstract:** Skyscrapers are symbols of local landmarks, and their prevalence is increasing across the world owing to recent advances in architectural technology. In Korea, the Lotte World Tower, which is now the tallest skyscraper in Seoul, was constructed in 2017. In addition, it has an observatory deck called Seoul Sky, which is currently in operation. This study focuses on the design of Seoul AR, which is a mobile augmented reality (AR) tour application. Visitors can use Seoul AR when visiting the Seoul Sky Observatory, one of the representative landmarks of Seoul, and enjoy a 360° view of the entire landscape of Seoul in the observatory space. With Seoul AR, they can identify tourist attractions in Seoul with simple mission games. Users are also provided with information regarding the specific attraction they are viewing, as well as other information on transportation, popular restaurants, shopping places, etc., in order to increase the level of satisfaction of tourists visiting the Seoul Sky Observatory. The final design is revised through heuristic evaluation, and a study of users' levels of satisfaction with Seoul AR is conducted through surveys completed by visitors to the Seoul Sky Observatory.

**Keywords:** mobile augmented reality tour application; Seoul Sky Observatory; interaction design

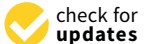



## 1. Introduction

### 1.1. Research Background and Goal

A skyline refers to the overall outline created by buildings in the center of a city and serves as a crucial element in determining the landscape and impression of a city. Skyscrapers are landmarks that represent countries and contribute to improving an urban area's image [1]. The construction of skyscrapers requires advanced technology and must also take into account numerous conditions with respect to a country's capital, the existing infrastructure of cities, social consensus, and relevant policies. As skyscrapers can be used to symbolically show off their regional or national capability and competitiveness, many countries have competitively built increasing numbers of skyscrapers in their major cities [2]. Moreover, skyscrapers are regarded as cultural and economic symbols of a city, and they are used as major tourism resources based on their symbolic characteristics as landmarks of the city [1]. Skyscrapers have received much attention from tourists as urban symbols, and observatories in skyscrapers serve as tourism destinations that enable visitors to view the entire landscape of a city. Observatory spaces in skyscrapers function as important tourist attractions and cultural places where various observation facilities and products are provided to help visitors increase their understanding of the general and cultural aspects of the city [3].

The Lotte World tower, which is a representative skyscraper in Seoul, South Korea, was opened in April 2017. The Seoul Sky Observatory, which operates on floors ranging from Floors 117 to 123 in this skyscraper, is located 500 m from the ground; it is the highest observatory in South Korea and the fifth highest observatory in the world. This observatory provides a 360° view of the entire landscape of Seoul and features various facilities and content designed to attract visitors to Seoul [4].

In this study, we aimed to develop an application that provides visitors to the Seoul Sky Observatory with information on the characteristics of this observatory, enabling them to view the entire landscape of Seoul, as well as amusing and important information. This app is also intended to ultimately increase the satisfaction of visitors to this observatory. To this end, the application Seoul AR was designed; this is a mobile AR application that can be used in the Seoul Sky Observatory. In addition, the satisfaction of visitors to this observatory was analyzed using this app.

*1.2. Major Contributions*

For a review of the existing research, this study examined the cases and characteristics of tour applications that apply mobile AR. Specifically, we carried out an inspection of the characteristics of the contents provided in the observatories of representative skyscrapers in Seoul, as well as of the contents provided in major skyscrapers in the United States and Japan. Moreover, to gain a clear understanding of the Seoul Sky Observatory, apart from an initial visit to the observatory for the purpose of observation alone, this study conducted in-depth interviews with 10 visitors to the Seoul Sky Observatory. The subjects were in their 20s to 50s, as most visitors to the observatory are in this age group, and the subjects experienced no problems with using smart devices. Therefore, the individuals within this age group were expected to use Seoul AR. Accordingly, the interview data were analyzed to identify the needs of visitors to the Seoul Sky Observatory and to design the requirements for Seoul AR.

Based on the analysis results, a scenario was developed for Seoul AR and an initial prototype of this app was developed. Seoul AR provides visitors to the Seoul Sky Observatory, a landmark of Seoul, with information about Seoul while they enjoy a 360° view of the entire landscape. In the development process, the app was designed using Sky Friends characters to provide a sense of consistency and unity with the content that already existed in the observatory. The users can identify a tourist attraction of interest in Seoul while playing simple mission games. They are also provided with information regarding the specific attraction as well as other information on transportation, major restaurants, shopping places, etc. Subsequently, the heuristic evaluation of this prototype was carried out by three experts who had experience in the development of mobile AR applications. This study made adjustments according to the design problems found in the evaluation process and reflected such changes in the final prototype design of Seoul AR. Furthermore, we conducted a service satisfaction evaluation of the visitors to the Seoul Sky Observatory to help us design the final prototype of Seoul AR. In addition, the limitations and expected implications of this research are discussed.

Seoul Sky is a skyscraper observatory in Lotte World Tower, and it is considered a major landmark in Seoul. Different types of observatories are in operation in high-rise buildings in several countries, such as the United States and Japan, and they differ in terms of their operation and the ways in which they accommodate visitors and their activities. As there is no existing case where an AR application is used for operations in skyscrapers, this study is expected to serve as an interesting example of AR application design in the Seoul Sky Observatory for researchers who wish to explore AR applications in this area. When comparing spaces that support various AR services, the process of designing an AR application for use in an observatory space in a skyscraper is rare. The issues that have been identified through trial and error during the design process are thought to be helpful not only to the study's research team but also to other researchers who plan to develop AR applications for observatory spaces in the future.

We expect that both researchers and readers will find the findings of our study to be helpful as an example of mobile AR application design.

## 2. Previous Works

### 2.1. Mobile AR Tour App

Information in the tourism industry has a higher value than that in other industries. For this reason, it is expected that tourism services that apply information and communications technologies (ICTs) will be utilized more widely in order to develop the tourism industry [5]. For example, ICTs can be used to provide visitors with service information such as the promotion of tour destinations, food, transportation, reservations, and tour guides for visitors conveniently and effectively [6]. In particular, the range of app services based on smartphones and ICTs has been expanded in our daily lives through the development of smartphones, which are equipped with cameras for taking photographs of surrounding objects, global positioning systems (GPSs) to inform users about their location, various sensors for detecting position or brightness, high-resolution displays, high-performance processors with enhanced graphic processing capabilities, and advanced high-speed wireless communication technology that enables fifth-generation (5G) and wireless fidelity (Wi-Fi) connectivity. Such technological advancements have established an appropriate technical environment for providing mobile AR services [5,6].

Mobile AR refers to a technology that applies three-dimensional (3D) virtual information to images or backgrounds in reality by using mobile devices such as smartphones. It combines the real world with the virtual world in real time to create an image based on both worlds and to augment object or image information [7]. Forecasts indicate that by 2025, the mobile AR market in South Korea will record an average annual growth rate of 31.1%. In particular, it is expected that mobile AR services will be developed mainly in the travelling, tourism, education, and shopping industries [8].

Pokémon GO, a location-based mobile AR game developed by Niantic in 2017, is a representative case that shows the successful spread of mobile AR services. The global success of this game enabled users to naturally experience mobile AR content [9] and led to the introduction of various mobile AR services in various markets.

In South Korea, location-based mobile AR games that are similar to Pokémon GO (e.g., CatchMon developed by MGAME, Seoul Catcher developed by HanbitSoft, and Joke developed by Hyundai Card) were released in the market after the successful launch of Pokémon GO [10–12]. Cities also utilized location-based mobile AR technology to develop both game and mobile tour AR content related to regional economy and tourism. For example, Yongin City in South Korea developed a mobile tour AR app based on a Cong Almon game to encourage regional tourism and economic revitalization. This app is installed in smart devices used by visitors, such as smartphones and tablets, and provides them with tourism and travelling information relating to ancient palaces, relics, historic sites, leisure, and camping based on the intuitive and effective use of AR [13,14].

Seoul City in South Korea also offers a mobile AR app service called Palace in My Hand, which provides information on four main palaces (i.e., Gyeongbokgung palace, Deoksugung palace, Changgyeonggung palace, and Changdeokgung palace) and the Jongmyo shrine located in this city. Visitors can learn detailed information through an audio guide by allowing their smartphone cameras to recognize markers placed at these sites [15,16]. Deoksugung Palace in My Hand, which is an app belonging to the Palace in My Hand app series, provides tourists with information about the Deoksugung palace and 130 interesting cultural properties located in Jeongdong, an area adjacent to the palace, based on AR content [17]. In addition, this app includes AR pathfinding content, AR content showing palaces and cultural properties in the form of 3D models, and travelling content based on object recognition technology, enabling users to experience relevant cultural assets in the past, the present, and the future.

In a similar case, Japanese researchers developed a mobile AR app called "take a walk" at the M Navi AR Ruins to provide tourism-related content for the Minami-Alps city in Yamanashi prefecture of this country. This app implements AR content on invisible underground ruins, and users can utilize such content to look around the city center and find relics in the area [18]. When the app recognizes markers installed in the city center, users

can see ruins through the cameras in their smartphones and listen to audio information. They can use this mobile tour AR app to naturally take a tour of the city, including underground relics; this tour will also teach them the history of key sites in the city and provide information about ruins [19].

Besides tourism purposes on land, there are also apps such as Maritime Educational Trip (MET), which combines geolocation information with knowledge of a specific location and presents it with AR under the sea [20].

There are also examples of the use of AR for navigation. A mobile navigation app that supports the activities of users during both normal times and disasters was developed by integrating augmented reality and web geographic information systems using landmark icons and pictograms as AR markers [21]. Additionally, a mobile app called Augmented Reality Tourism System (ARTS) uses 3D scans converted from a point cloud to a portable interaction size to display interactions between cultural heritage, tourism, and pedagogy [22]. In addition, a marker-free system was also proposed for AR-based indoor navigation. The proposed system uses an RGB-D camera to observe the surrounding environment and builds a point cloud map using Simultaneous Localization and Mapping (SLAM) technology. It is expected that this will be applied for smart glasses and that smartphones could be used as touring tools [23].

Other examples of mobile tour AR apps include an app called the House of Olbrich, which introduces the architectural history of Darmstadt in Germany; an app called Krka National Park Tour, which provides information on the Krka national park in Croatia based on AR; and a mobile tour AR app, which shows information on buildings at the University of Washington based on mobile AR [24–26]. The characteristics of the aforementioned mobile tour AR apps are as follows.

1.  They show necessary information based on AR graphics and voice assistance provided by the smartphones of users without the use of guidebooks to increase the understanding of users.
2.  They display information according to the location of users and the geographical properties of the space. For example, as shown in the case of the World Around Me (WAM) app, they overlap actual background with information on important places for users, such as automated teller machines (ATMs) and public toilets, based on the user's location; they can also make use of the camera view functions of smartphones to provide users with convenient navigation [27]. Apps such as Virtlo, which offers information on restaurants and shopping areas near users and relevant coupons based on navigation and travelling information, have contributed to the promotion of regions and growth in regional economies [28].
3.  Users can learn the history of tourist cities or key sites in a more engaging and amusing way by naturally moving to the places introduced in these apps and directly experiencing these places based on storytelling.

### 2.2. Content for Observation Deck of Skyscrapers in Seoul

An observation deck, observation platform, or viewing platform is an elevated sightseeing platform that is usually situated upon a tall architectural structure, such as a skyscraper or observation tower [29].

This study selected as research targets the tourist attractions of the Seoul Sky Observatory, the N Seoul Tower Observatory, and the 63 Art Observatory, which are the top three observatories in terms of size and popularity among all the observatories of skyscrapers in Seoul. This decision was made in accordance with Visit Seoul Net (visitseoul.net), an official website for tourism information operated by the city. Accordingly, we investigated the content provided for visitors to these observatories [30].

1.  "Seoul Sky Observatory" (height: 500 m above sea level):

The Seoul Sky Observatory is part of the Lotte World tower, which is located in Jamsil, Seoul, and operated by Lotte. It is located at the 117th to 123rd floors of this tower. Media

art content focusing on the themes of traditional Korean culture and the Lotte World tower as a landmark of South Korea are provided on the basement and entrance floors. The observatory deck of the Seoul Sky Observatory consists of the observatory area, which enables visitors to view the entire 360° landscape of Seoul; the Sky Deck, which has a transparent floor; the Magic Deck, which uses polymer dispersed liquid crystal (PDLC) screens to make its opaque glass floor seem transparent; and observation facilities in the Sky Terrace, an outdoor observatory area. The content provided in this observatory includes simple games played on media tables, which help game participants learn information about this landmark, and regular music performances in observation areas [4].

2. "N Seoul Tower Observatory" (height: 480 m above sea level):

The N Seoul Tower Observatory is part of the N Seoul Tower, which is located in Namsan, Seoul, and operated by the CJ Group. This observatory includes a Shocking Elevator, which shows videos based on a special display installed on its ceiling during its upward and downward operation, and a Shocking Wall, which shows videos based on a projection device installed on its upper area for visitors in the waiting space prior to entering the Shocking Elevator. On the observatory deck, visitors can communicate with visitors at the Pusan Tower Observatory (height: 289 m above sea level) by using real-time interactive content on a device called the Connecting Tower. In addition, an audio guide service is provided at the windows of the N Seoul Tower Observatory to audibly inform visitors about the places shown through the windows [31].

3. "63 Art Observatory" (height: 264 m above sea level):

The 63 Art Observatory is part of the 63 Square Tower located in Yeouido, Seoul, and is operated by the Hanwha Group. Located on the 60th floor of the 63 Square Tower and combined with an art exhibition space, this observatory has been operated throughout constant changes since 2008 [32]. Observatories in Japan, such as the Sky Tree Observatory (height: 480 m above sea level), the Sky Circus Observatory (height: 251 m above sea level), the Tokyo Tower Observatory (height: 223 m above sea level), and the Roppongi Hills Mori Tower Observatory (height: 250 m above sea level), also operate various content, including art exhibitions, animations, and VR experiences, based on different themes to attract tourists [33–36]. Since the opening of the 63 Art Observatory, 33 special exhibitions, including an exhibition called Kitty S. Art and Design Exhibition Seoul, which was held in 2008, and an exhibition called TEAMBOTTA63 BOTANIC EFFECT, which was held in May 2021, as well as 50 mini exhibitions have been arranged. This observatory has also operated events where artists who are both exhibitors and visitors can communicate with each other. Furthermore, this observatory consists of the Sky Tunnel, which is connected to the opposite side of this observatory and decorated with colored lighting; the Wall of Wishes, to which visitors can attach pieces of paper including their written wishes; and the Thrill Deck, which applies optical illusion effects based on mirrors to make the observatory floor appear transparent [32].

The One World Observatory (height: 380 m above sea level) is part of the One World Trade Center, which was built in New York, the U.S., in May 2015. This center is located at the site of the former World Trade Center, which was destroyed by terrorists on 11 September 2001. The One World Observatory not only provides observation areas but also content on the history of Manhattan through the application of advanced digital technology. The Top of the Rock Observatory (height: 259 m above sea level) is an observation deck of the Rockefeller Center that shows historical events related to the observatory in the form of visual media [37].

As shown in the aforementioned cases, observatories in skyscrapers across the world are utilizing new media technology and content to introduce historical events related to these observatories and reflect their image as a symbol of cutting-edge technology. In other words, these buildings are being transformed into spaces of comprehensive content that provide visitors with various types of entertainment and experiences beyond the basic function of allowing visitors to view the landscape of the city [38]. Moreover, observato-

ries across the globe are being developed as entertainment spaces beyond their original role as tourist attractions, such as by providing virtual reality content according to their characteristics, as in the case of the Tokyo Bullet Flight and Swing Coaster VR programs, which can be experienced in the Sky Circus in Japan [30]. On the other hand, the Seoul Sky Observatory, the N Seoul Tower Observatory, and the 63 Art Observatory, which are located in representative skyscrapers in Seoul, mainly provide content introducing and promoting these locations.

## 3. Designing Seoul AR for Seoul Sky Observatory

### 3.1. Understanding of Seoul Sky Observatory

The Seoul Sky Observatory is located in Jamsil, Seoul. On a clear day, it offers a panoramic view of the adjacent Gangnam area as well as the central area of Seoul. On the way from the ticket office to the elevator going up to the observatory floor, various exhibition contents are displayed, as shown in Figure 1d. The Sky Shuttle elevators (Figure 1a), which hold the official Guinness World Records for the two categories of "The longest distance traveled via double-deck elevator" and "Fastest double-deck elevator", travel 496 m at 10 m/s.

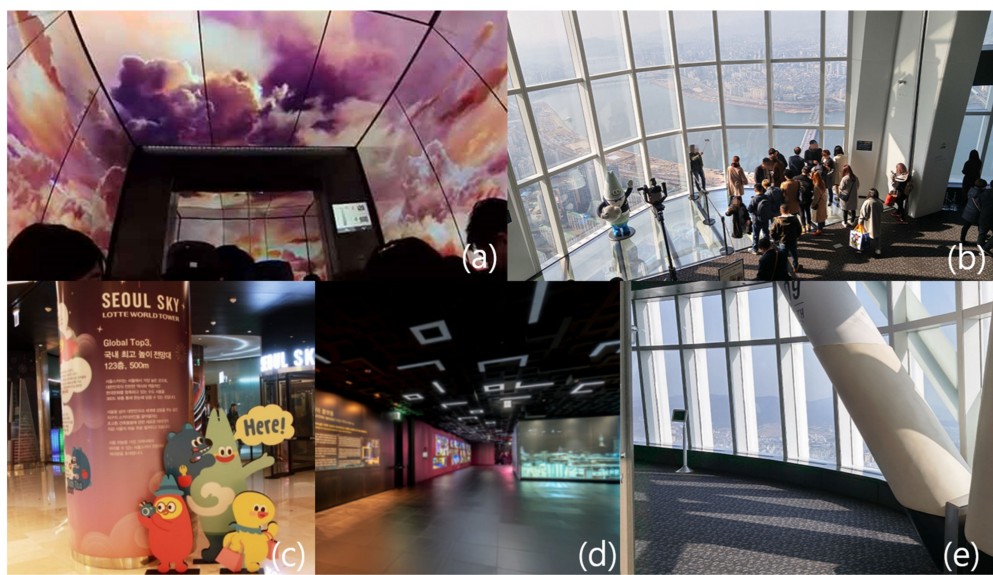

**Figure 1.** Seoul Sky Observatory: (**a**) Inside the Sky Shuttle Elevator, animations relating to Seoul and the Lotte World Tower are displayed. (**b**) On the Sky Deck on the 118th floor, you can appreciate the height of the observatory by looking through the glass floor. (**c**) At the entrance to Seoul Sky, you can see various Seoul Sky characters: Pix Lolo, Yum Tao, Tower Lota, and Chu Tete (from left). Chu Tete is the character used in the Seoul AR application in this study. (**d**) The platform for the Seoul Sky elevator is connected through spaces with various media displays. (**e**) From the 118th floor of the Seoul Sky Observatory, a 360° view of the entire landscape of Seoul can be enjoyed.

During the approximately 1 min ride in the elevator on the way to the observatory floor, the changing views of Seoul are shown in animation through the five-sided display built into the elevator. The place that draws the most people in Seoul Sky is the Sky Deck (Figure 1b), built on the 118th floor at a height of 478 m, where one can see the bottom floor through transparent glass; hence, the location is popular, with many people wishing to experience the height. People also flock to the Sky Terrace, the outdoor space, where one can enjoy the view with the naked eye from the 120th floor. In the evening, many people come to watch the sunset and the city landscape at nighttime. The cafe on the 122nd floor and the restaurant on the 123rd floor are also packed with people. There is no limit to the length visitors can stay in the observatory throughout the day; hence, visitors are at leisure to take pictures in the observatory spaces, such as the Sky Deck or Sky Terrace. There are couches all over the observatory space on the 118th floor (Figure 1e), and people

can sit comfortably while watching the view. The visitors are of diverse age groups, but the majority of them are in their 20s to 50s.

Along with the observation, this study involved conducting in-depth interviews with 10 people who visited the Seoul Sky Observatory at least once to develop observatory content based on mobile AR and to analyze the backgrounds of visitors to this observatory, such as their need for and satisfaction in the content provided at this location. Since the interview was conducted before the development of the app, the content of the interview questions only covered the experience of the visit to Seoul Sky Observatory, as well as visitors' expectations and disappointments and the main activities they experienced during the visit. In-depth interviews are a useful means for investigating the opinions and feelings of users. This study carried out in-depth interviews with those who visited the Seoul Sky Observatory to identify their backgrounds, including their motives for visiting this observatory and their needs for necessary information and content. We also intended to obtain information that could not be gained through quantitative research, such as visitors' hidden desires, attitudes, and emotions [39].

The interview participants who were selected were in their 20s to 50s. The reason for the selection was because, during the preliminary observation, it was determined that this age group was representative of most of the visitors to the observatory. In addition, since people in this age group had more experience using smartphones compared to other age groups, we decided that it is more likely that they would be more likely to use the AR app; thus, they were selected for the interview. Interview questions focused on the following topics: (1) their motives and expectations prior to visiting the observatory, (2) their experience and impressions during their visit, and (3) their expectations regarding re-visiting this observatory, as well as their opinions on the content provided at the observatory and their satisfaction or dissatisfaction with the use of this observatory after their visit. To effectively compare and analyze the interview data, standardized open-ended interviews were implemented. Specifically, each interview participant was asked interview questions along the same theme. However, certain questions were added or excluded according to the participants' experience. Each interview was conducted for approximately 20 min and recorded for analysis after agreement was received from the interview participant.

### 3.2. Results of User Interviews

The data of the interviews were summarized, focusing on opinions mentioned two or more times in a similar context. Table 1 shows the interview result. As indicated in this table, 4 out of 10 interview participants commented that the Seoul Sky Observatory provided insufficient levels of entertainment in consideration of the long waiting time, expensive entrance fees, and observation facilities provided.

**Table 1.** Details of interviews with visitors to the Seoul Sky Observatory and their satisfaction with this observatory.

| Code | Sex (Age) | Interview Scene | Important Contexts |
|:---:|:---:|:---:|:---:|
| A | F(26) |  | A long queue caused discomfort. Problems included the long waiting time, low accessibility, and absence of content that visitors could enjoy except for the observation facilities in the observatory. |
| B | M(56) |  | The audio guide service provided in this observatory was impressive because it introduced the main places in Seoul, which were seen through each window, while visitors viewed the entire landscape of the city. |

**Table 1.** *Cont.*

| Code | Sex (Age) | Interview Scene | Important Contexts |
|------|-----------|-----------------|--------------------|
| C | M(32) |  | This observatory focused on souvenir shops and restaurants rather than observation contents despite the high entrance fees. |
| D | F(27) |  | The landscape of Seoul as viewed from this observatory was impressive. However, it was difficult to identify information relating to specific objects and areas. |
| E | F(30) |  | It was unsatisfactory that explanations of the landscape were not provided. The satisfaction of visitors could be increased if the observatory provided explanations of the history of Seoul or the main events held in this city in a place where visitors could view the urban landscape. |
| F | M(24) |  | The underground floor through which visitors passed prior to entering the observatory floor was impressive. The overall displays installed in the high-speed elevator were also impressive. The observatory floor is an ideal place for using observation facilities and taking photos that will be uploaded to social media. |
| G | F(32) |  | It was satisfactory for visitors to view the landscape of Seoul as soon as they arrived at the observatory floor. However, the method employed to find places and objects in this observatory was different from that based on the map. For this reason, it was difficult to determine what was located in each direction based on the building icons. |
| H | F(53) |  | The entrance fee was high and visitors were not provided with sufficient services that they could enjoy, even though this observatory consisted of several floors. |
| I | M(29) |  | This observatory could be interesting for those visiting Seoul for the first time. However, the experience is not impressive overall owing to the long waiting time as well as the expensive entrance fee. There is limited content for visitors to participate in or enjoy inside the observatory. |
| J | M(33) |  | It is impressive that the opaque observatory deck becomes transparent. People may find the night view impressive if they visited the observatory at night. Unlike the news on an air base located near this observatory and relevant flight safety, it was difficult to find this place from the observatory deck. |

Two interview participants expressed positive opinions regarding the services and facilities based on the view. However, the entire set of interview participants stated that this observation focused on theme-based facilities such as souvenir shops and restaurants rather than observatory decks and that they were provided with insufficient content that enabled them to view and appreciate the landscape. Several interview participants discussed their expectations prior to their visit to the Seoul Sky Observatory, saying that they expected to broaden their understanding of Seoul through their visit to this landmark of Seoul.

However, counter to their expectations, they were not provided with sufficient information about Seoul in this observatory. Moreover, they found it difficult to identify the location or direction of the main buildings pointed out, even though they were allowed to view the entire landscape of Seoul on the observatory deck. They also expressed that they would be interested in receiving information on the history of Seoul or the main events held in this city. The analysis of the expectations and disappointments of the interviewed visitors at Seoul Sky showed that there was insufficient content for visitors to enjoy inside the observatory, which they entered after waiting a long time and paying an expensive entrance fee. Moreover, there were insufficient options offered during the visit, other than the observatory facility itself, leading to a short and uneventful visit experience. Based on the findings, to increase the length of stay and the level of satisfaction of visitors to Seoul Sky, in this study, the authors developed an application named Seoul AR that provides location-based guidance to major tourist attractions, buildings, and historical places in Seoul using the observatory space.

Seoul AR utilizes the features of the Seoul Sky Observatory to allow visitors to observe the entire area of Seoul. Users can use their smartphones to locate specific tourist attractions they wish to find in the observatory space; detailed information on the places that can be seen from the observatory is also provided. In particular, considering the various age groups of the visitors, the app provides more than just information on places of interest. Users of the app can accomplish small missions, such as finding an attraction in the observatory space using a smart device; for each mission, they will acquire information step by step. Consequently, the design of the app aims to increase visitors' length of stay in the observatory and users can find various well-known tourist attractions of Seoul through an intuitive and fun method.

The user interface and user experience (UI/UX) in AR are based on constant interactions with users, unlike existing 2D-based static interfaces. For this reason, the AR-based UI/UX can change depending on the environment or conditions. Thus, an essential requirement for AR apps is to help users recognize information relating to their location efficiently and conveniently [40]. This study analyzed the needs of users of the mobile AR tour app for the Seoul Sky Observatory and the AR and UI/UX design elements required for this app based on interview data to enable this app to provide users with essential information in an efficient and convenient manner and to determine design elements for this app. Table 2 describes the details of such needs and the required elements for this app.

**Table 2.** Design elements defined for Seoul AR.

| Items | Details |
|---|---|
| Needs of users | Content making use of the characteristics of the observatory to allow an extensive view of the entire area of Seoul is needed. As a landmark of Seoul, information that is relevant to Seoul sightseeing (such as its tourism spots and history) needs to be provided. In addition to the observatory facility, other content should be provided that users can experience and enjoy in the observatory. |
| Characteristics and elements of Seoul AR | Location-based service—various types of information should be provided according to the location of users in the observatory. A user-driven system—users should be allowed to select necessary information for themselves. Interactive storytelling—the app should provide an interesting experience, such as by encouraging users to seek interesting places, solve quizzes, and visit locations. The app should also effectively inform users about relevant information. |

**Table 2.** *Cont.*

| Items | | Details |
|---|---|---|
| Mobile UI/UX elements for Seoul AR | Logical UI | This app should help users find actual spaces that they would be willing to visit and provide them with relevant information in the observatory. |
| | | This app should allow users to obtain information relating to the location of each place by playing staged games. |
| | Graphical UI | Text consistency, visibility, and aesthetic aspects should be considered in this app. |
| | | Graphic consistency and aesthetic aspects should be considered in this app. |
| | | Multimedia consistency and conciseness should be considered in this app. |
| | Information UI | Consistency and efficiency related to methods of seeking places based on navigation and the providing of information should be reflected. |
| | | The efficiency of the methods used for seeking correct places and storing information should be reflected. |
| | | Useful information should be provided on different places. |
| | UX | Simple missions to places that users are willing to visit should be developed to provide them with the pleasure of visiting these places in an entertaining way. |
| | | Useful information on target places should be provided to increase the satisfaction of users. |
| | | This app should be designed to help visitors to the observatory utilize it in a convenient and amusing way to increase their satisfaction in visiting this observatory. |

### 3.3. Seoul AR Design

Seoul AR is a mobile AR tour app that can provide visitors to the Seoul Sky Observatory with important information regarding their visit to Seoul in an entertaining way by considering the characteristics of this observatory that enable them to view the entire landscape of Seoul. The Seoul AR app helps users to locate a place in Seoul that the user wishes to visit by viewing the location from the Seoul Sky Observatory and efficiently providing a description of the place, along with useful information about tourism and transportation. To help users find relevant information about the place of interest in a fun and active way, game elements have been incorporated into the app—for instance, the user can complete missions and obtain information about the locations seen as well as other detailed information about the observatory.

In this study, we decided to develop this app to increase the satisfaction of visitors to the Seoul Sky Observatory. Visitors can view a 360° landscape of Seoul from the Seoul Sky Observatory, and landmark icons showing the main buildings and locations in Seoul are placed below each window to help visitors pick out these buildings and places from the corresponding directions, as shown in Figure 2. However, the visitors we interviewed found it difficult to visually identify target places based on these landmark icons owing to the absence of relevant information. This was because they did not know the exact direction or distance where buildings were situated, and they did not know the shape of the building or place concerned.

Considering the aforementioned environment of the Seoul Sky Observatory, mobile AR was adopted in Seoul AR to more easily provide information relating to the main tourist attractions in Seoul for visitors to this observatory. With respect to the concept of this app, simple games were developed to help users seek desirable places by themselves and utilize this app in a more entertaining way. Moreover, Sky Friends, which are symbolic characters for the Seoul Sky Observatory, as shown in Figure 1c, were used to form a friendly and consistent impression for the benefit of visitors to this observatory.

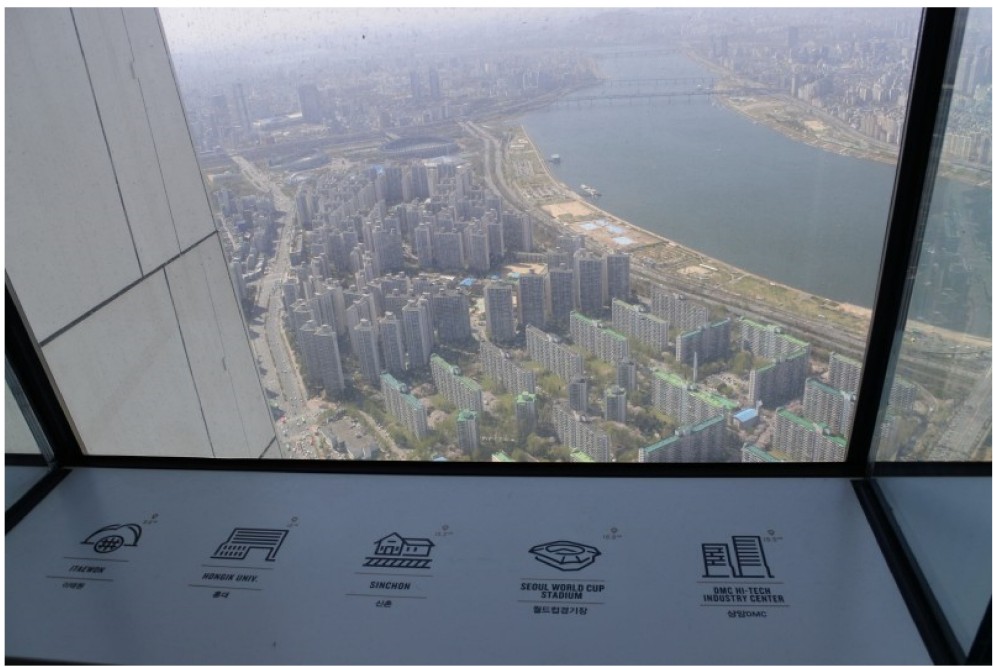

**Figure 2.** Visitors can view a 360° landscape of Seoul from the Seoul Sky Observatory; landmark icons for the main buildings and places in Seoul are placed below each window to help visitors pick out these buildings and places from the corresponding directions at the 118th floor.

As Seoul AR is an app that is used at the Seoul Sky Observatory, the lines and colors of the existing representative characters and character designs of the Seoul Sky Observatory have been employed to maintain a sense of unity and consistency with the other content available at the observatory.

As indicated in Figure 3, a scenario was designed so that the Seoul AR user would move across the observatory space in the direction of the tourist attraction they wish to visit. Subsequently, the attraction can be viewed; by playing simple game missions using the AR display, the actual location of the attraction in relation to the observatory can be found and the user can save the description of the place of interest as well as the information related to tourism, transportation, and shopping.

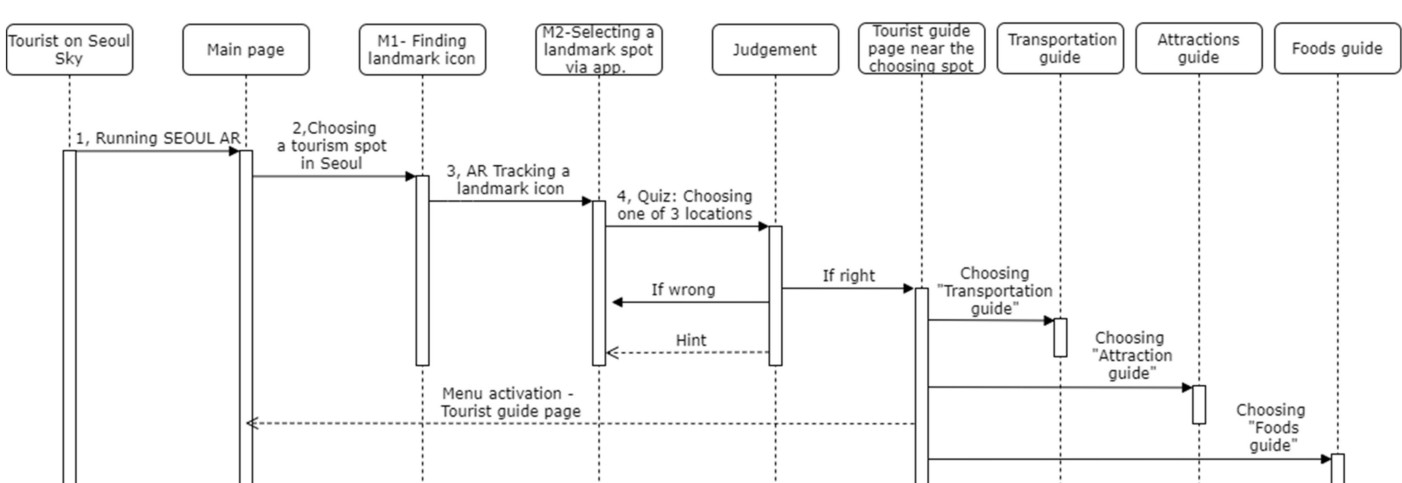

**Figure 3.** Seoul AR scenario flows.

The Samsung Galaxy Note 8 based on the Android operating system and the Apple iPad Air 2 based on IOS were utilized to develop Seoul AR. Unity 5.6.0f2 (64 bit) and the SDK 6.2 version of Vuforia for Unity were used to implement the AR.

Fifty landmark icons that were used to indicate the main places in Seoul in each location from the Seoul Sky Observatory were formed through graphic work and applied as markers, as shown in Figure 2.

After operating Seoul AR, users move to find the landmark icon displayed on the screen in the second stage. When they move their smart devices to the target landmark icon, this app recognizes the actual landmark icon by referring to the image target stored in the Vuforia Database. When the landmark icon is successfully recognized, the on Tracking Found method is used on Canvas. Subsequently, star-shaped AR objects, which are established in advance on image targets in the direction of the corresponding window from the target market, are displayed on the screen.

The screen configuration of Seoul AR allows users to obtain tourist information about Seoul while performing their missions one at a time. In terms of screen design, instead of showing too many images and too much text information on one screen, Chu Tete, one of the observatory characters, was actively used on the screen display so that anyone could accomplish these missions step by step, facilitating the easy and intuitive use of the app.

As shown in Figure 4, users can select a category from among several categories related to nature, ruins, and culture on the home screen of Seoul AR. After selecting the category, as shown in Figure 5, the user is given their first mission to find the landmark icon of the corresponding place, after which the user moves in the corresponding direction by finding the landmark icon that matches that displayed on the screen in the observatory to complete Mission 1. The user can see the circle when they succeed in this mission.

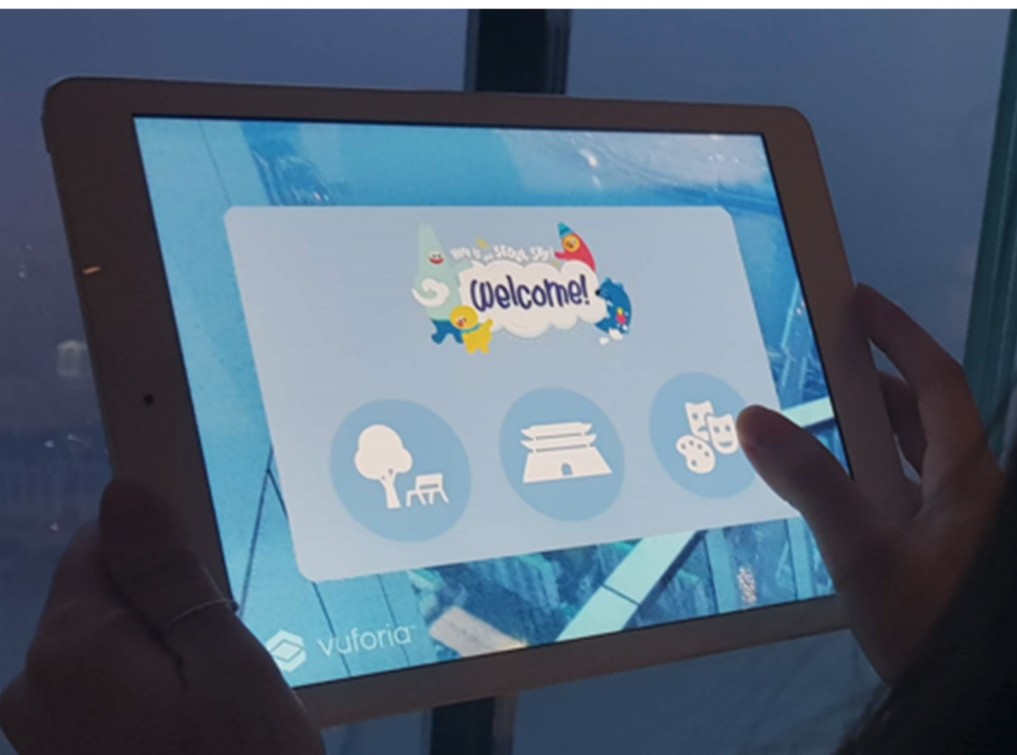

**Figure 4.** On the home screen, a user can select one of the categories of tourist attractions that can be viewed from the observatory: nature, ruins, or culture.

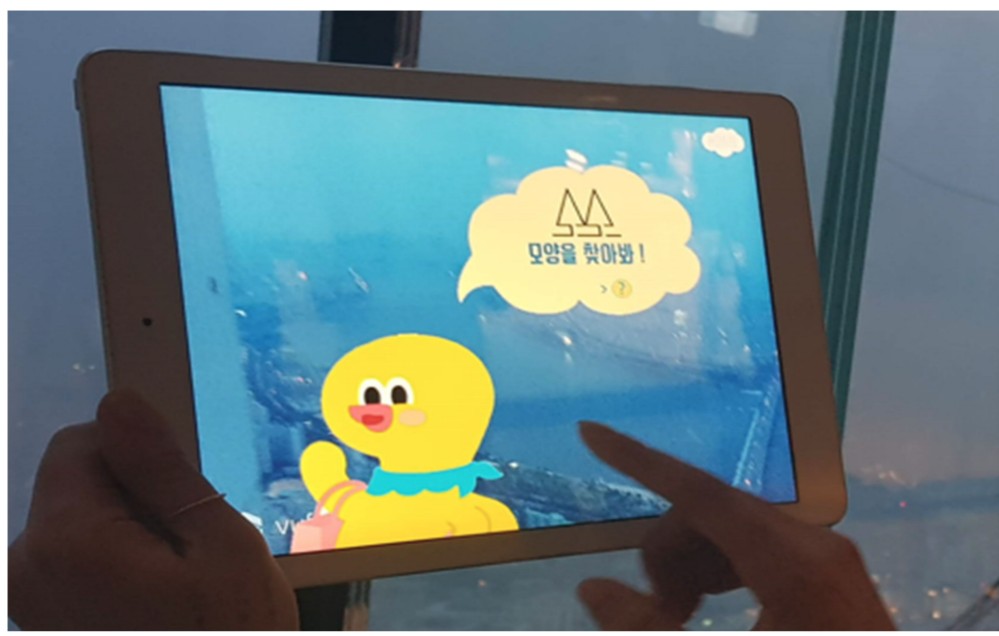

**Figure 5.** Users need to find the landmark icon in the observatory space that matches one of the tourist attraction categories they have selected.

Subsequently, the user can move the camera on their smart device to make this app recognize the corresponding landmark icon and pass to the second stage, as shown in Figure 6. For Mission 2, the user needs to place the camera on their smart device in the direction of a window, below which the corresponding landmark icon is marked.

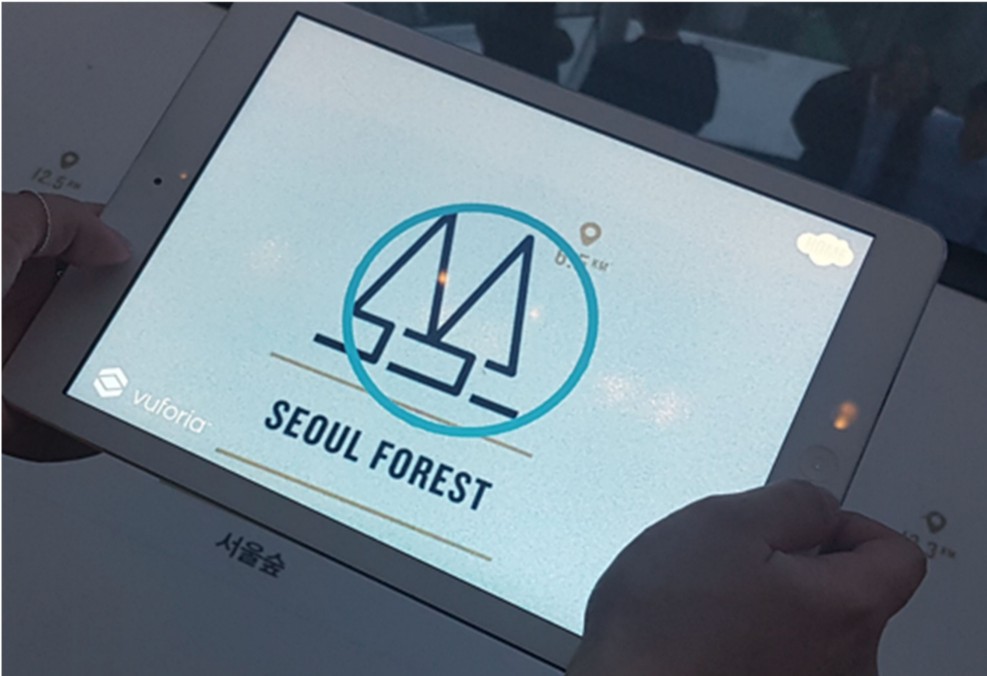

**Figure 6.** Mission 1 is completed when users find the landmark icon they are looking for in the actual observatory space and register the icon through the app.

Subsequently, the screen displays three star-shaped AR objects, including an AR object indicating the correct location of the place related to the corresponding landmark icon. Users select the correct location of the place related to the landmark icon based on their decisions. If the correct location is selected, as shown in Figure 7, users can

see the circle when they are successful and receive information on the corresponding tourist attraction, such as general guide information, transportation, and restaurants, as shown in Figures 8 and 9. Although they perform Missions 1 and 2 only once to find the target location, this app provides them with information on locations related to other landmark icons. Based on this scenario, users can easily obtain information on Seoul tourist attractions that can be observed from the Seoul Sky Observatory without constantly having to complete missions.

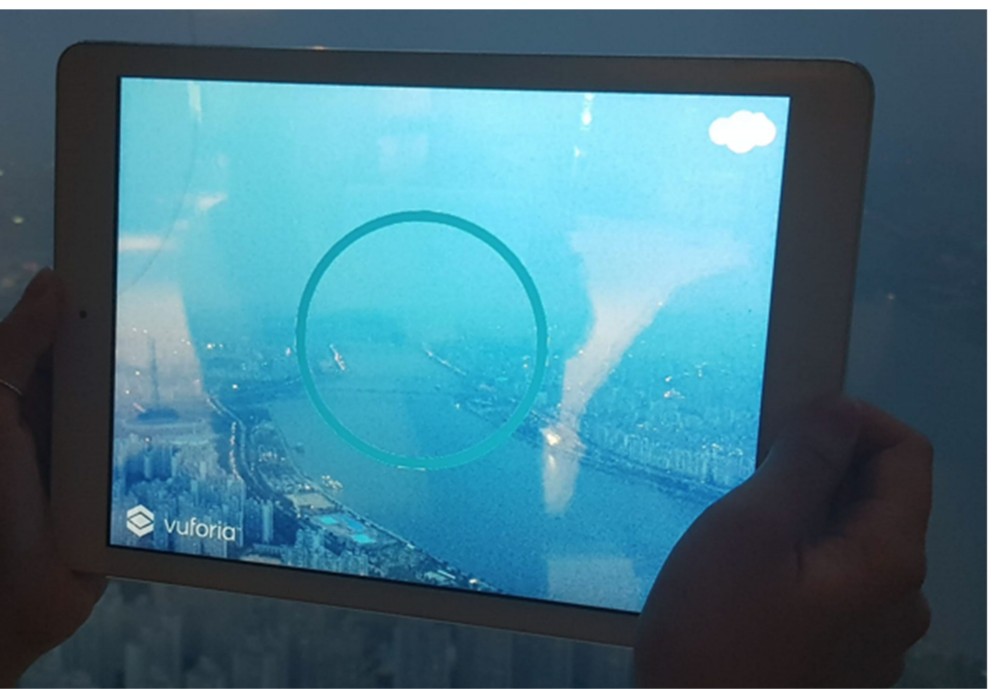

**Figure 7.** After successfully completing Mission 1, Mission 2 can only be completed after finding the actual place in the observation space; then, detailed information about the location is provided to the user. The user can see the circle when they are successful.

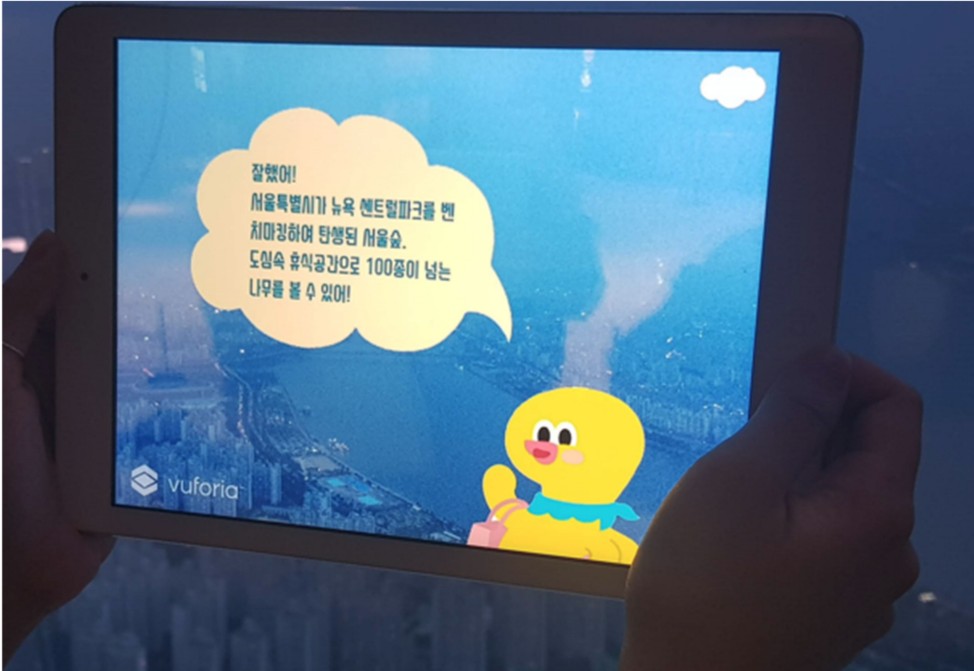

**Figure 8.** After successfully completing Mission 2, the user can see a brief description of the relevant place.

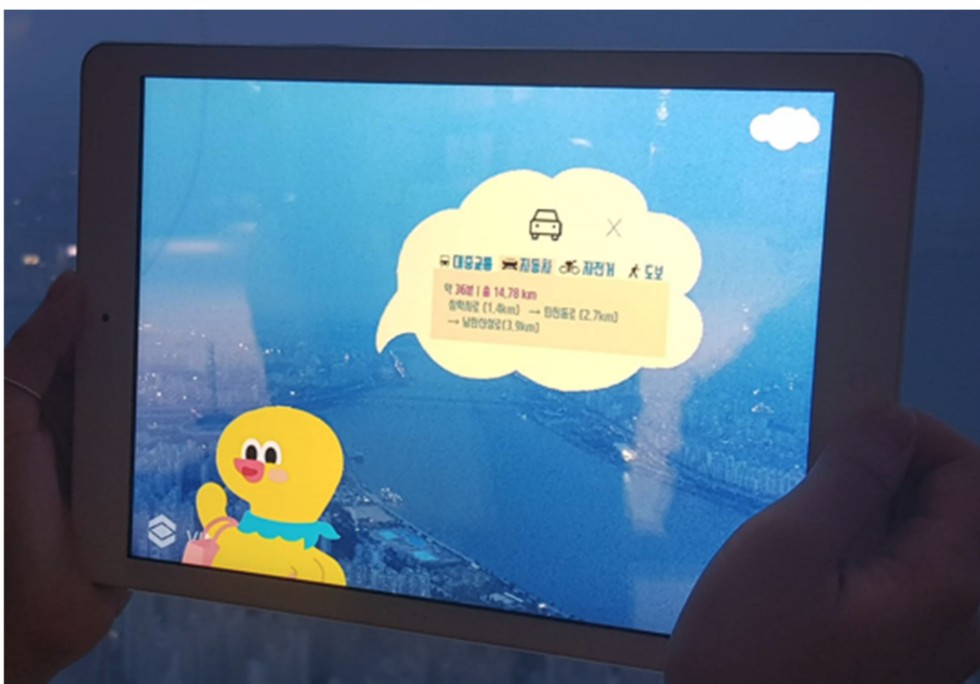

**Figure 9.** Users can view traffic information, tourist information, and information regarding recommended restaurants near the location.

## 4. Evaluation Seoul AR for Seoul Sky Observatory

### 4.1. Design Feedback

This study made use of various opinions of mobile AR application experts to improve and supplement the design of the initial Seoul AR prototype. Heuristic evaluation was used to identify the representative problems in terms of the usability of the designed AR prototype application.

Heuristic evaluation was first proposed by Nilsen. As it has the advantages of providing a fast evaluation, a low cost, and insight into a target system, it is often used as a cost-effective method for evaluating the usability of early prototypes [41]. Three experts who had prior experience in the use of AR and the development of mobile AR apps were selected as participants in the heuristic evaluation. They consisted of two UI/UX designers with five years of experience and a mobile AR app programmer with three years of experience.

In the selection of experts, three people with experience that is highly relevant to the research subject were finally selected from the list of experts recommended by our colleagues. The evaluation process was performed by arranging individual meetings with the experts. It was difficult to invite them to the Seoul Sky Observatory owing to scheduling issues, among other problems. Therefore, the researchers visited the workplace of the experts. Interviews were conducted individually, and each interview took approximately 30–90 min. Prior to the interview, informed consent was obtained so that we could proceed with recording.

As the evaluation results differed between the experts, for the preliminary evaluation criteria, the 22 principles for AR-based usability proposed by Ko (2013), based on Nilsen's usability evaluation principles, were utilized. Twenty-two principles for AR-based usability (i.e., availability, consistency, context-based, defaults, direct manipulation, enjoyment, error management, exiting, familiarity, feedback, help and documentation, hierarchy, learnability, multi-modality, low physical effort, navigation, predictability, personalization, recognition, responsiveness, user control, and visibility) were established and provided in the form of a table to help us evaluate the developed AR app. This helped us to better consider

the application of AR technology in this app and solve the problem of our insufficient objectivity, which was highlighted as a heuristic evaluation problem [42].

To this end, an orientation was conducted prior to the heuristic evaluation to enable the experts to understand the initial Seoul AR prototype and the established usability principles in advance, as well as to help them evaluate the developed app based on their understanding. The experts were asked whether they had sufficient experience using the Seoul AR app. The experts were requested to identify problems with the current design based on the 22 principles established for AR-based usability, and interviews were conducted with them.

The analysis was conducted using card sorting based on the 22 principles for AR-based usability table completed by the experts and on the recorded interview data. Consequently, 10 problems were identified based on the expert evaluation, including some problems that were mentioned twice. Table 3 shows the problems that were identified.

**Table 3.** Findings from expert reviews of the early prototype of Seoul AR.

| Items | List of Questions Identified | Design Adjustment | Usability Elements |
|---|---|---|---|
| Menu movement and navigation | Initial setting for halting tasks or returning to previous conditions of tasks required difficult processes according to the stage. | A home button was added to each stage on the screen. | Navigation |
| | When it was difficult to complete a mission, the corresponding stage was maintained. In addition, a stage for preventing the discontinuity of the mission on the app was not provided. | Tutorials were included. A function for providing a hint in the event of failing to complete a mission was also added. | Help and documentation, error management |
| Screen design | A more aesthetic design, including color selection, was required to provide a pleasant experience for users. | A character from the Seoul Sky Friends was incorporated into the screen design. The color of the character was also reflected in the screen design. | Enjoyment, visibility |
| Information readability | Readability was reduced owing to the large amount of text on buttons according to categories. | The font size was changed from 14 pt to 16 pt. | Recognition, visibility |
| | Information on tourist attractions—readability was reduced because information on the sub-stages of the stage related to information on transportation was only indicated on one page. | Repeated keywords were replaced with pictograms. A large amount of information was provided over several pages. A menu function for providing information on the different landmarks and tourist attractions was activated after a mission was successfully completed. | Hierarchy, visibility, feedback |
| | Information on tourist attractions—readability was reduced because information on the sub-stages of the stage related to information on popular restaurants was only indicated on one page. | | |

**Table 3.** *Cont.*

| Items | List of Questions Identified | Design Adjustment | Usability Elements |
|---|---|---|---|
| Fun and interest related to the use of the app | The level of interest was reduced because the game missions were too easy. | The level of game missions increased in stages in consideration of the various age groups using the developed app. The characters of Seoul Sky Friends were used to increase the interest of users, and feedback on sound generated from manipulation was reflected. | Consistency, feedback, responsiveness |
| | The directions of AR objects in the app were slightly different from the location of actual landmarks. | Foot-shaped stickers were attached to locations where users could use their mobile devices to cause the developed app to recognize landmark icons, which were 1 m away from the center of windows, including the icons that were on the ground. | Low physical effort, user control |
| | As the text was provided in a literary style, it appeared static and reduced the level of interest. | The text was provided in a colloquial style and mission stories were developed based on communication with a character. | Familiarity, user control |
| | The level of interest was reduced because of the large number of characters used. | One character was used, and the representative color and shape lines of the character were incorporated into the app design. | Familiarity |

*4.2. Design Revision for the Final Prototype*

Ten problems identified through the heuristic evaluation were classified as relating to "menu movement and navigation", "screen design", "information readability", and "fun and interest related to the use of the app". Design adjustment solutions were developed to solve each problem, and the final design was derived based on these solutions. Table 3 shows the details of these solutions.

As each user showed different levels of interest in information related to Seoul, categories according to places based on tourist attractions and shopping were generated instead of the previous method of selecting landmark icons randomly on the initial screen. Places were located based on the user's proximity to the Seoul Sky Observatory. The initial screen design, which had been based on the four characters of Seoul Sky previously, was changed to only use the Chu Tete character to reduce visual confusion. In particular, it was found to be necessary to reduce the error between the location of AR objects and the location of actual landmarks, which was caused by users having different heights and differences between their standing locations and the directions of markers, as explained in Figure 10.

*4.3. Evaluations*

This study evaluated the service satisfaction of visitors to the Seoul Sky Observatory, when using Seoul AR to determine the effect of the use of the final prototype of Seoul AR on the satisfaction levels of visitors to this observatory, using the questions that are included in Table 4.

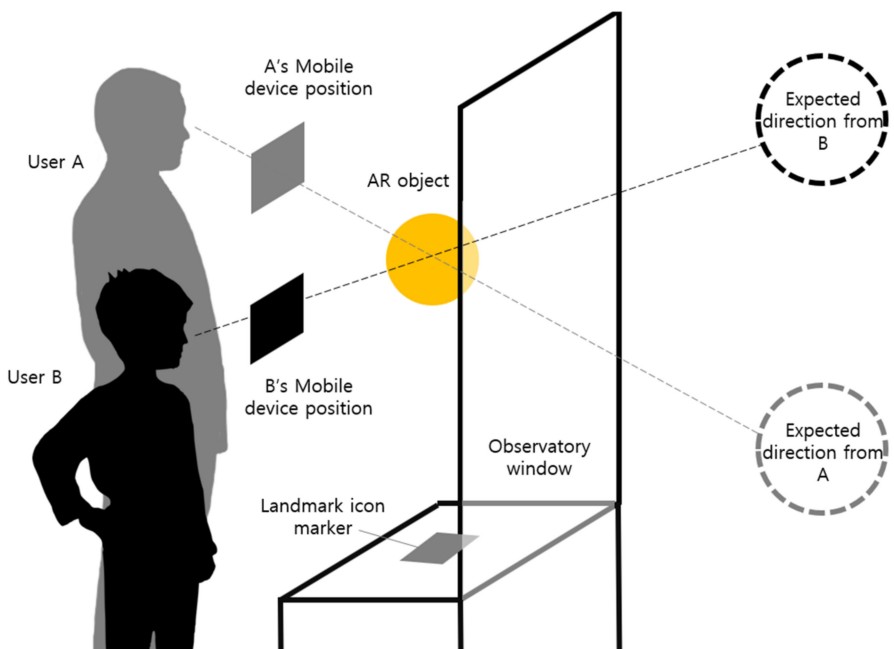

**Figure 10.** It was found that, depending on the height of the user, the height at which they used their smartphone, and the location of the user, there was a point where the direction and position of the place, as described by the AR object shown to be the actual landmark icon marker, were different from the real place. Therefore, we tried to reduce the error by using the app 1 m away from the average height of the main users and marker [43].

**Table 4.** Survey questionnaires on the satisfaction of users with using Seoul AR.

| Evaluation Factors | Survey Items |
| --- | --- |
| Q1—Service for tourist attraction information | Q1-1 Are you satisfied with Mission 1 provided in the developed app as a service for providing tourist attraction information? <br> Q1-2 Are you satisfied with Mission 2 provided in the developed app as a service for providing tourist attraction information? <br> Q1-3 Are you satisfied with the brief introduction of tourist attractions provided in the developed app as a service provided for tourist attraction information? <br> Q1-4 Are you satisfied with the information on tourist attractions, such as information relating to tours, transportation, and popular restaurants, provided in the developed app as a service for providing tourist attraction information? <br> Q1-5 Did you find the service for tourist attraction information provided by the developed app to be helpful for increasing your satisfaction with visiting the observatory? |
| Q2—Service for the observatory | Q2-1 Are you satisfied with Mission 1 provided in the developed app as a service for the observatory? <br> Q2-2 Are you satisfied with Mission 2 provided in the developed app as a service for the observatory? <br> Q2-3 Did you find the service for the observatory provided by the developed app to be helpful for increasing your satisfaction with visiting the observatory? |
| Q3—Service for entertainment | Q3-1 Are you satisfied with Mission 1 provided in the developed app as a service for providing entertainment? <br> Q3-2 Are you satisfied with Mission 2 provided in the developed app as a service for providing entertainment? <br> Q3-3 Are you satisfied with the information on tourist attractions, such as information relating to tours, transportation, and popular restaurants, provided in the developed app as a service for entertainment? <br> Q3-4 Did you find the service for entertainment provided by the developed app helpful in increasing your satisfaction with visiting the observatory? |

**Table 4.** *Cont.*

| Evaluation Factors | Survey Items |
|---|---|
| Q4—Searching for tourist attractions | Q4-1 Are you satisfied with the process used for seeking places that you are willing to find based on Missions 1 and 2 provided by the developed app? Q4-2 Did you find the function of seeking places provided by the developed app helpful in increasing your satisfaction with visiting the observatory? |

In the survey on the satisfaction of visitors based on the utilization of Seoul AR, the service satisfaction elements were classified as satisfaction in obtaining information on tourist attractions (service for tourist attraction information), satisfaction in using the service that provides information on the landscape (service for the observatory), satisfaction in using the entertainment-related service provided by the developed app (service for entertainment), and satisfaction in seeking and confirming actual locations by themselves (searching for tourist attractions). Fourteen survey questions according to each element were generated and measured based on a five-point Likert scale.

The survey participants were selected among visitors to the Seoul Sky Observatory. Specifically, 36 females and males who were in their 20s to 50s, who had visited the Seoul Sky Observatory at least once, and who experienced no difficulty in using smartphones were selected as the final survey participants.

Prior to the evaluation, the participants were asked to sign forms of agreement for having the actions they undertook while using Seoul AR and their survey responses recorded. During the orientation session, they were provided with a brief introduction to Seoul AR, its usage methods, and its evaluation methods. With respect to the Task session, they joined Missions 1 and 2 based on a landmark icon by using Seoul AR and experienced the flow of searching relevant tourist information on the target tourist attraction. As soon as they completed these missions, they participated in surveys. They freely expressed their opinions on using Seoul AR during executing the tasks, and a survey was conducted for approximately 10–20 min per participant.

### 4.4. Results of Evaluation

To evaluate the satisfaction of users when using Seoul AR, 36 participants (average age: 31.9 years old) in their 20s to 50s were selected. The proportion (52.8%) of 19 female participants was slightly higher than that (47.2%) of the 17 male participants, as shown in Table 5. With respect to the age groups of participants in the usability evaluation of the developed app, those in their 20s accounted for 58.3% of all the participants, those in their 30s accounted for 22.2%, those in their 40s accounted for 13.9%, and those in their 50s accounted for 5.6%. Based on this result, the ratio of participants in their 20s was higher than that in other age groups.

**Table 5.** Information of the evaluation participants.

| Sort | Details | Number | % |
|---|---|---|---|
| Gender | Male | 17 | 47.2 |
| | Female | 19 | 52.8 |
| Age group | 20s | 21 | 58.3 |
| | 30s | 8 | 22.2 |
| | 40s | 5 | 13.9 |
| | 50s | 2 | 5.6 |
| Frequency of visits | Once | 30 | 83.3 |
| | Twice | 2 | 5.6 |
| | three times or more | 4 | 11.1 |

With respect to the frequency of visits to the Seoul Sky Observatory in the population, the usability evaluation of participants who had visited this site once accounted for 83.3% of the entire participants, thus showing the highest ratio. Those who had visited this site

twice accounted for 5.6%, and those who had visited this site three times or more accounted for 11.1%. Based on the survey result, the participant data that were regarded as noise were excluded. Consequently, the data of 35 participants were analyzed.

### 4.4.1. Q1 Service for Tourist Attraction Information

The level of user satisfaction with the service for tourist attraction information based on Mission 1, which requires users to move to landmark icons, was evaluated to be the lowest, as shown by the result of M = 2.82 (SD = 0.57) indicated in Figure 11, where the *X*-axis represents the type of question asked and the *Y*-axis represents the five-point Likert scale.

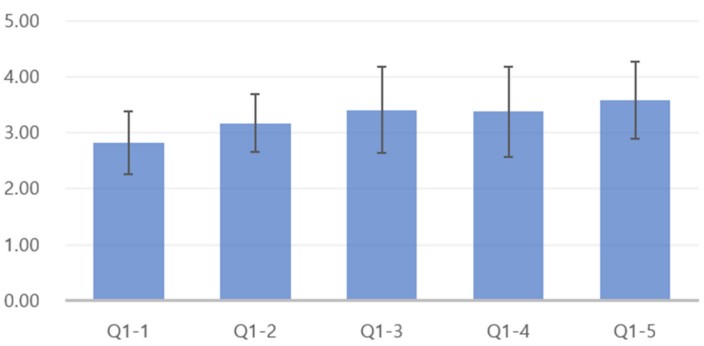

**Figure 11.** Results showing service satisfaction with tourist attraction information.

The satisfaction with the service provided for tourist attraction information based on Mission 2, which requires users to find the actual locations of target places based on the developed app, was evaluated to be lower than the mean values for other types of satisfaction, as shown in the result of M = 3.18 (SD = 0.51). However, the level of satisfaction with the service for tourist attraction information related to the brief introduction of tourist attractions, a tour, transportation, and popular restaurants, and information provided by the developed app was higher than the medium value, as shown by the results of satisfaction with the service for tourist attraction information of M = 3.41 (SD = 0.77), M = 3.38 (SD = 0.80), and M = 3.59 (SD = 0.69).

This result indicated that users tended to show higher levels of satisfaction when obtaining tourist attraction information comprising a brief introduction of tourist attractions, a tour, transportation, and popular restaurants after completing missions when compared with their satisfaction when obtaining such information while seeking landmarks and the actual locations of these landmarks.

Given that users could not receive tourist attraction information immediately through Mission 1, they tended to be more satisfied with the tourist attraction information received through Mission 2, which relates to finding actual locations using AR objects after recognizing landmark icons. This was attributed to the fact that users were not sufficiently aware of the sequence and progress of the app; however, the app's purpose and the ways to navigate it were explained to users before the experiment.

### 4.4.2. Q2 Observatory Service

With regard to questions relating to the effect of the service provided by Seoul AR on user satisfaction when visiting the observatory, the user satisfaction with service for the observatory based on Mission 1 was calculated to be M = 3.53 (SD = 0.55), and that based on Mission 2 was calculated to be M = 3.79 (SD = 0.76), as shown in the Q2 service result for the observatory indicated in Figure 12, where the *X*-axis represents the type of question asked and the *Y*-axis represents the five-point Likert scale. The satisfaction of

users with the service for the observatory based on this app was calculated to be M = 3.71 (SD = 0.67), and thus, was higher than the medium value.

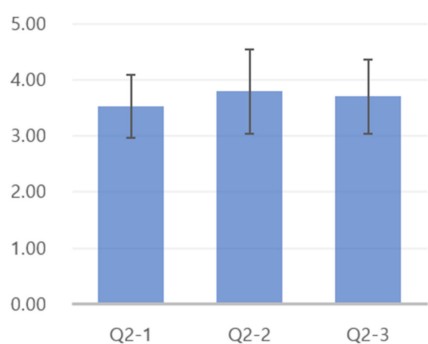

**Figure 12.** Results showing user satisfaction with the observatory service.

This result indicated that users found Seoul AR to be helpful in providing services for the observatory, and that they also considered the process around their participation when finding the direction of tourist attractions based on the app and landmark icons in the observatory space to be helpful for providing services for the observatory.

For Mission 1, users need to recognize places in the observatory from where they can have a 360 degree-view of Seoul in addition to the observatory facilities such as the Sky Deck and Sky Terrace, which are typically very crowded. The app prompted users to look for landmark icons while moving in Mission 1 and to make observations that allowed them to find the corresponding tourist attractions in Mission 2. This was more interesting for the users and they tended to be satisfied with this new content. To complete Mission 1, the users had to find corresponding icons among 50 landmark icons arranged in 360 degrees in the observatory and recognize them through their camera. Hence, the users found the process of completing Mission 1 to be easier and more satisfactory than Mission 2, which required them to find the locations of tourist attractions at relatively fixed places. Consequently, improvements were made to the "hint" feature, which informs users of the corresponding locations when they fail to recognize landmark icons in Mission 1.

4.4.3. Q3 Entertainment Service

With regard to questions relating to user satisfaction with the entertainment service provided by Seoul AR, the level of satisfaction based on Mission 1 was calculated to be M = 3.91 (SD = 0.56), meaning that this was higher than the medium value. Based on Mission 2, it was calculated to be M = 3.88 (SD = 0.76), which was also higher than the medium value. These results are indicated in the result for entertainment service in Figure 13, where the *X*-axis represents the question asked and the *Y*-axis represents the five-point Likert scale.

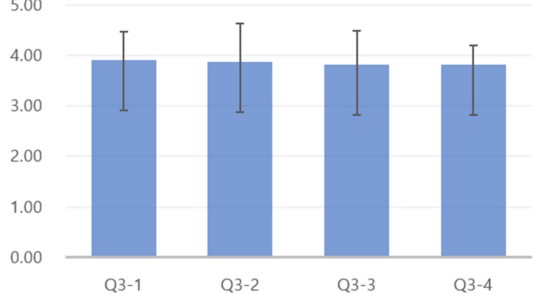

**Figure 13.** Results showing user service satisfaction with the entertainment services.

The level of satisfaction with the service provided for tourist attraction information, including a brief introduction, a tour of the location, and information about transport and popular restaurants, was calculated to be M = 3.82 (SD = 0.66), and the satisfaction with the entertainment service provided by the developed app was calculated to be M = 3.82 (SD = 0.38). Thus, both values were evaluated to be high overall.

This result indicated that users regarded the processes of completing missions, moving to tourist attractions, and receiving information about tourist attractions, such as a brief introduction and tour based on Seoul AR, as entertainment services, and that they tended to be satisfied with these services.

Users had generally seen the view of Seoul from several places in the Seoul Sky Observatory, and this app allowed users to find the directions to tourist attractions from the observatory, where they could see the entire city, in Mission 1. Then, in Mission 2, users could directly locate the places seen from the observatory and obtain information about these places in addition to tourist information. This app provided an opportunity for users to try to find the locations and information of tourist attractions they were looking for in a more interesting manner compared to their past experiences in other observatories, where they just enjoyed the panoramic view of Seoul and then descended from the tower.

### 4.4.4. Q4 Searching for Tourist Attractions

With regard to the service provided for searching for tourist attractions based on Seoul AR in the Seoul Sky Observatory, the mean value of satisfaction based on Missions 1 and 2 was calculated to be M = 3.5 (SD = 0.65), and that of satisfaction upon finding tourist attractions based on the developed app in the observatory was calculated to be M = 3.74 (SD = 0.74). Thus, these mean values were considered to be positive, as shown in the result for Q4, "Searching for tourist attractions", in Figure 14, where the *X*-axis represents the question asked and the *Y*-axis represents the five-point Likert scale.

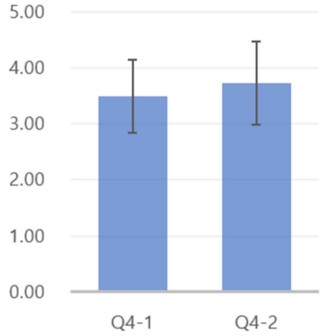

**Figure 14.** Results showing user service satisfaction with searching for tourist attractions.

This result indicated that users found the use of Seoul AR partially helpful for finding tourist attractions that they were interested in finding in the observatory space.

The users tended to think that the introductions to the tourist attractions they were looking for and the tour-related information provided through Missions 1 and 2 helped them to find information about tourist attractions. Upon the completion of one mission, the users received all the available tourist information about a location without having to perform other missions, which implies that it was easy to check the tourist information of other landmark icons using this app. Therefore, it was assumed that searching for tourist information through this app was more satisfying than finding it through completing Missions 1 and 2.

## 5. Discussion and Conclusions

This study designed Seoul AR to provide visitors to the Seoul Sky Observatory, the highest observatory in Seoul, with information on the main tourist attractions in this city, as well as transportation options and popular restaurants, in a convenient manner.

Seoul AR is a mobile AR tour application that is based on smart devices, including smartphones, which enables users to identify the direction of a tourist attraction that they are interested in finding by using landmark icons placed below windows at the Seoul Sky Observatory as markers. Users can also find the actual location of the tourist attraction observed from the corresponding direction by completing simple missions and can receive a brief introduction to the target place, a tour, and information on transportation options and popular restaurants in the area.

Moreover, a heuristic evaluation was performed by three experts with experience in the development of mobile AR apps. The initial design of Seoul AR was adjusted based on the evaluation result and these changes were reflected in the process of developing the final prototype design. To evaluate the ultimate Seoul AR prototype, interviews were conducted with 36 visitors to the Seoul Sky Observatory. Specifically, service elements were classified as services for providing tourist attraction information, observatory services, and entertainment services to help us analyze the effect of Seoul AR on the satisfaction of those who visited this observatory. Accordingly, the service satisfaction of visitors was quantitatively evaluated according to the elements classified.

The results gained from analyzing the satisfaction of users of Seoul AR were as follows. First, the users expressed a positive level of satisfaction, when visiting the Seoul Sky Observatory, upon the completion of missions, provided in Seoul AR, that involved seeking the direction of a tourist attraction of interest and finding the actual location of this site based on the landmark icons indicated in the observatory. They also tended to be satisfied with the ability of this app to provide observatory services and entertainment services.

With regard to the tourist attraction information provided by the app, which received a lower score than the other service items, it was found that this app should be updated to increase the number of options for tourist attractions that users can select, subdivide information on tourist attractions, and improve the tourist attraction information offered. In particular, voice assistance, which was not included in the current version of Seoul AR, was noted as a requirement, given that this app is used based on smartphones.

In addition, based on the observations of the process by which users used Seoul AR, the following limitations of the current version of this app were identified.

First, there was a resolution problem when this app was used on a smartphone. As this app was developed for use on a tablet device, the resolution of this app on a tablet was different from that on user smartphones. When users used this app on smartphones, the Chu Tete character was displayed on the screen with a relatively larger size, making the screen appear smaller. As graphic elements such as text also appeared larger on this app compared with those on general apps, simpler graphics and display compositions were noted as requirements.

Second, the visibility range was limited owing to the height of skyscrapers and the effect of weather. It is difficult to visually identify many of the tourist attractions located on the ground (with the exception of those located closer to skyscrapers) because of the characteristics of skyscrapers. Users were easily able to find the location of tourist attractions based on landmark icons, which served as markers during the process of completing Mission 1. On the contrary, they encountered several limitations when finding the correct direction of tourist attractions based on the developed app during the process of completing Mission 2. As the tourist attractions appeared much smaller from the skyscraper, users encountered difficulties when receiving explanations about tourist attractions and relevant information from the developed app under certain conditions, such as at nighttime, when tourist attractions were unlikely to be identified with the naked eye, or in an environment where the visibility range was not ideal, owing to obstacles such as fine dust.

Third, the presence of different physical user attributes led to problems when using this app. As users have different heights and exhibit different behaviors when using smart devices, the fixed location of AR events was often different from the actual location of the users. Although the locations of spots at which users were required to place themselves were marked, the majority of users were not aware of these marks. While the locations of AR events, which were designed in advance, were intended to help users find approximate directions, the degree of the support actually offered varied according to the different heights of the users and the different directions and locations of their smartphones.

The aforementioned problem was identified in advance through heuristic evaluation, and changes based on the problems identified were reflected to mark the locations of spots at which users should place themselves and adjust the AR events based on the average height and eye level of males and females with ages ranging from their 20s to 50s. Nevertheless, several users performed the motion of spreading their fingers on the center of the screen when a general camera app was opened in order to zoom in on the screen, owing to the problem of the smaller screen image of the actual location.

Further research on the application of machine learning for the training of landscape images from target locations will be conducted to help develop an enhanced mobile AR tour app. This app will be able to provide the location of major tourist attractions more accurately regardless of the different heights of users, use the real locations of users when they seek the locations of tourist attractions from the Seoul Sky Observatory, and inform users about the actual location of the tourist attractions that they are interested in finding in a more convenient manner.

**Author Contributions:** Software, S.S.; writing—review and editing, S.S. and Y.C.; supervision, Y.C. All authors have read and agreed to the published version of the manuscript.

**Funding:** This work was supported by the Sogang University for Yongsoon Choi (Research Grant 201510001.01).

**Institutional Review Board Statement:** This study is based on the 1st author's master's work and includes human subjects. However, the user's personal identification information used in this study did not include personal information other than age and gender information. Thus, ethical review and approval were not required for the study on human participants in accordance with the local legislation and institutional requirements.

**Informed Consent Statement:** Written informed consent has been obtained from the interview and user study participants to publish this paper.

**Data Availability Statement:** The data presented in this study are available on request from the corresponding author.

**Conflicts of Interest:** The authors declare no conflict of interest.

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
