# Peer review of "SEOUL AR: Designing a Mobile AR Tour Application for Seoul Sky Observatory in South Korea"

_electronics, doi:10.3390/electronics10202552_

Round 1

Reviewer 1 Report

Pls find the following review comments:

1. Does `AR' stand for `Augmented Reality'?

There are many undefined notations in their 1st appearance. 
For examples: N Seoul Tower.

2. Fine-tune the text in some parts of the paper!

-Hence needs through proofread.

3. There are many unnecessary  small paragraphs. Reduce the no. of paragraphs by merging few paragraphs into one
as deem fit.

4. Figure 4 consists of 4 sub-figures. Hence at first separate them considering  4(a), 4(b) etc.

Use the figure caption at the bottom of each figure, rather than on top. undefined Y-axis!

5. Research range and methods--->Major contributions

Emphasize major contributions in a concise manner in the abstract and an elaborative manner in the section, `1.2 Major contributions'.

6. `1.2 Research background and goal' needs improvement.

7. Presentation of the paper needs improvement.

Author Response

 Firstly, we are grateful for your detailed feedback.

  1. Yes, we confirm that “AR” stands for “Augmented Reality.” Since the term is used repeatedly in this paper, “Augmented Reality” is abbreviated as “AR” in line 5 of page 6, which is the first time the word is used in the main text. N Seoul Tower is the name of a building with an observatory in Seoul, the capital of South Korea. To reduce this type of confusion, double quotation marks (“”) have been placed around proper nouns, such as “N Seoul Tower.” And the name and place of the building are capitalized.
  2.  We will make sure that the manuscript undergoes proofreading and editing before the final publication.
  3. We have merged some of the smaller, unnecessary paragraphs into the main relevant one to reduce the number of paragraphs.
  4. Following your comment, in Figure 5, each graph was numbered and classified accordingly. Further, the Y-axis is the scale value of the Likert scale. We inserted this information before the text where the contents of the graph in Figure 5 (current Figure 11) are described to reduce confusion in page 29.
  5. Some of the revised content in Section 1.2 is now reflected in the Abstract.
  6. The content in Section 1.2 has been revised.
  7. In addition to the background and goal of this study above, we tried to reduce ambiguity in the contents of observations and user interviews in the Seoul Sky observatory before the design session. And we tried to explain the design part of the Seoul AR application in detail. Afterward, the heuristic evaluation part was also edited to explain about 22 AR usability principles and one figure was added to explain the problem of the prototype and design feedback. For the presentation, we followed the journal format and balance was considered when placing figures and tables. We adjusted the size and position of the figures, table, and paragraphs in detail and also applied the style of the caption. We also merged some of the smaller and paragraphs to the main relevant to reduce the number of paragraphs. 

Reviewer 2 Report

This study developed an AR application for use in the observatory by analyzing the needs of tourists, and used questionnaires to understand user satisfaction, which has certain application value.

This study can be considered for publication, but it is recommended that the authors increase the description of the system development process and system interface to provide readers with more references.

Author Response

We are grateful for your thorough feedback. We have provided more details on the description of the system development process, and additional explanation has been provided for the parts that were difficult to understand. The content of the system development process and system interface has been revised on pages 10–15.

Reviewer 3 Report

This paper describes the design and evaluation of an augmented reality app to assist in visiting the Seoul Sky Observatory.

The authors engaged in user interviews to understand the users' needs and pain points when visiting the observatory, performed an heuristics evaluation over an initial prototype of the app, and a final user study to find the satisfaction when using the app.

On a broad level, I think the authors followed a good design approach, with an initial attempt at understanding needs, and then developing and evaluating two prototypes. However, each phase does not seem to have been developed in much depth.

However, scientifically, I see little contribution. This paper describes the results of an interaction design work process. What are the scientific contributions? This is yet another app/yet another usability evaluation. How can the results be generalized to other situations? What do readers learn?

More specific comments

  • The observatory should be described in more detail. What can be seen from there? How do users behave there? What do they go there to see? The paper shows only on photo of the location and little description of the movement and behaviour of users.
  • The need for the proposed app should be clearer. And the functionality of the app should be better described. What problem is the app trying to solve? What information exactly does it provide. The few screenshots are not enough to understand it. The design of the app should also be clearer. The paper mentions that one of the main difficulties users face is in identifying the buildings they can see from the observatory. But then the app features a game that helps users only indirectly. This is not explained. Why make it difficult for the users to identify the buildings?  Why a game. Also why use a duck character. The app's users are described as ranging from 20 up. So why an apparently childish design? Table 2 is not very clear. The details are rather vague: "spatial characteristics ... should be utilized", "useful information should be provided...", "content should be developed to encourage the participation". What spatial characteristics?, what useful information?, participation in what kind of activities? Also it is not clear how the table was derived from the interviews. How did interactive storytelling appear to be necessary? The app should be described in more detail. It is not even clear how or what AR overlays are shown to users... It is also not clear why the height of the user influenced the errors in placing the stars.
  • When describing the interviews it is not clear when they were made. "Interview participants were selected among those who were in their 20s to 50s and they were expected to use the mobile AR app developed in the Seoul Sky Observatory." Does this mean that the app was already made when the interviews were held? Or are the authors just describing the target audience for the app? What questions exactly were asked? Did participants consent in having their photos taken and published?
  • When describing the heuristics evaluation, more details would be important. What heuristics exactly were used. The paper shows only a list of 22 keywords, but these are not understandable as heuristics. Where did the evaluators conduct the analysis? They they go up to the observatory? How long did it take to make the analysis?

Author Response

We are grateful for your detailed review of this manuscript and the valuable feedback. This study was conducted while the first author was working as a media exhibition manager at the Seoul Sky Observatory in 2019. At that time, the research was conducted while developing an AR mobile application for use in the Seoul Sky Observatory to increase the level of satisfaction of the visitors.

We would like to submit this paper to the Special Issue "LifeXR: Concepts, Technology and Design for Everyday XR" of Electronics. This special issue has encouraged the submission of papers over a wide range of topics, such as user experience of mobile XR/ LifeXR, LifeXR applications, and LifeXR in everyday spaces, as well as design solutions to LifeXR; therefore, the authors consider the paper to be relevant to the topic.

As you have commented, this research investigates interaction design, which is different from the existing usual direction of Electronics, so we weighed on the possibility and relevance of the submission of this paper to the Special Issue "LifeXR: Concepts, Technology and Design for Everyday XR" of Electronics. After discussing with the person in charge of the journal, it was decided that the paper was pertinent to the themes covered in the Special Issue; thus, we have submitted the paper accordingly.

From a technical perspective, the contribution may not be novel because we have used Vuforia and Unity in this study, which are existing platforms commonly employed for AR mobile application development. However, in our opinion, the process of designing an AR application for use in the observatory space in a skyscraper is rare compared to the types of spaces supporting various AR services. The issues that have been identified through trial and error in the design process are thought to be helpful not only to the authors’ research team, but also to other researchers who plan to develop AR applications for observatory spaces in the future.

Seoul Sky is a skyscraper observatory in Lotte World Tower and it is one of the major landmarks in Seoul. As reviewed in this study, different types of observatories are in operation for high-rise buildings in several countries, such as the United States and Japan. These observatories have differences depending on the operation context and on how they accommodate the visitors and their activities.As there is no existing case of an AR application for operation in skyscrapers, we believe this study will serve as an interesting AP application design in the Seoul Sky Observatory for researchers who wish to explore an AR application in this area. The design process and method employed in this study may be difficult to generalize when developing AR applications in all types of observatories of various skyscrapers. However, for both researchers and readers, the findings of this study may still be helpful as a design example of a mobile AR application (referred to as Seoul AR), which was developed to enhance visitor satisfaction at the Seoul Sky observatory. 

The observatory should be described in more detail. What can be seen from there?

=> The content has been revised on page 14.

How do users behave there? ? What do they go there to see?

=>The content has been revised on pages 14-17.

The need for the proposed app should be clearer. And the functionality of the app should be better described

=> Based on this aspect, the content of the manuscript has been revised on pages 17–22.

What problem is the app trying to solve?

=> The content has been revised on page 17. Based on an analysis of the expectation and disappointment related responses in the interviews of Seoul Sky visitors, it was determined that there was insufficient content to make the visit inside the observatory enjoyable. This became a problem because the visitors had to pay an expensive entrance fee after a long wait, only to find out that the options offered for the experience, other than the observatory facility, were insufficient, making the visit short and uneventful.

Based on the findings, in this study, to increase the length of stay and the level of satisfaction of visitors to Seoul Sky, an application that the authors named Seoul AR was developed, which provides location-based guidance on major tourist attractions, buildings, and historical places in Seoul. Seoul AR utilizes the features of the Seoul Sky Observatory, which provides an extensive view across the entire area of Seoul. The users can use their smartphones to locate the specific spots they wish to find in the observatory space, and obtain detailed information on the places. In particular, considering various age groups of the visitors, the app provides generalized information on the place of interest and information for smaller missions, such as finding an applicable spot in the observatory space. They can use their smart device to acquire information on each step of the mission. Consequently, the design of the app aims to increase the visitors’ length of stay in the observatory, while the users experience various well-known tourist attractions of Seoul in an intuitive and fun method.

What information exactly does it provide.

=> The content has been revised on page 20. When a user selects one of the major categories of tourist attractions of interest in Seoul, i.e., nature, ruins, or culture, the place within the observatory, where the selected attraction can be viewed, is identified. Then, when the actual location is found through a simple game mission, a description of the place along with tourism information, including transportation, popular restaurants, and shopping places, are provided.

The paper mentions that one of the main difficulties users face is in identifying the buildings they can see from the observatory. But then the app features a game that helps users only indirectly.

=> The difficulty was identified during the user testing process after the app was designed. The user was able to find an approximate direction toward the place they wished to find using the developed app; however, owing to the nature of the observatory in a skyscraper, there were other variables such as weather. In addition, although this app was originally designed to provide tourist information, we observed that the users, while experimenting with this app, tried to manipulate the screen in search of a zoom function for the place they were looking for. Unfortunately, a zoom function is not provided in this version. It is indeed a key function that we would like to incorporate in the next version of the app. As described above, the main function of the game in the developed app is to allow the visitors of the observatory to find the tourist attractions they wish to visit in an easy and fun manner, to extend the length of stay, and to satisfy the search for new content in the observatory to make the visit enjoyable.

This is not explained. Why make it difficult for the users to identify the buildings? 

=>This content has been revised. We have added an explanation on page 18

why use a duck character

=>The content on this aspect has been added and revised on pages 14, 19, and 24 . In Seoul Sky, there are four existing characters called Sky Friends; initially, all the characters were used in the screen design. However, this was pointed out as a cause for confusion by the experts, who indicated a lack of unity in their assessment; therefore, the final design was based on the duck character, Che tete.

The app's users are described as ranging from 20 up. So why an apparently childish design?

=> The majority of visitors to Seoul Sky are in their 20s to 50s; hence, the main users were assumed to be in the 20–50 range. Moreover, since this age group has plenty of experience with smartphones and smart devices, the authors thought it would be suitable to consider them as the base for the main user of this app. The reason for the childish presentation of the overall design is that, although the main purpose of this app is to provide tourist information, rather than text-based information, the developed app aims for an easy and fun experience for the visitors of the observatory. In addition, since this is an app used within the Seoul Sky Observatory, we wanted to ensure that the content provided conformed to the rest of the content in the observatory, and hence, selected the representative character of the Seoul Sky Observatory along with the lines and colors of the character design.

Table 2 is not very clear. The details are rather vague: "spatial characteristics ... should be utilized", "useful information should be provided...", "content should be developed to encourage the participation". What spatial characteristics?, what useful information?,

=>The unclear details in Table 2 have been revised. The space of the Seoul Sky Observatory provides a 360° view of the entire landscape of Seoul; we referred to this aspect as spatial characteristics. In the revised manuscript, instead of unclear expressions, such as spatial characteristics, we have used more direct language and explicit expressions for clarity.

participation in what kind of activities? Also it is not clear how the table was derived from the interviews.

=> We have reduced ambiguous expressions and made corrections accordingly.

How did interactive storytelling appear to be necessary? The app should be described in more detail. It is not even clear how or what AR overlays are shown to users...

=> The explanation for the content of the app has been added and revised for each step in pages 18–22.

It is also not clear why the height of the user influenced the errors in placing the stars.

=>The description on this point was supplemented and revised on page 24; Figure 10 was also revised.

When describing the interviews it is not clear when they were made. "Interview participants were selected among those who were in their 20s to 50s and they were expected to use the mobile AR app developed in the Seoul Sky Observatory." Does this mean that the app was already made when the interviews were held? Or are the authors just describing the target audience for the app?

=>The description of this aspect has been supplemented in the manuscript. The interviews aimed to identify the needs of users who were visiting the Seoul Sky Observatory or those who had already visited the Seoul Sky Observatory. We also wanted to identify the areas that the visitors thought were different from their expectations. The interview participants were recruited mainly from people in their 20s to 50s. This is because, during preliminary observation, it was determined that most visitors to the observatory were within the age group of 20–50. In addition, since this age group has more experience in using smartphones compared to other age groups, they have a higher likelihood of actually using the AR app in development. Therefore, they were selected for the interview. Since the interview was conducted before the development of the app, the questions were regarding their experience on visiting the observatory as well as their expectations and disappointments.

What questions exactly were asked? Did participants consent in having their photos taken and published?

=> Each interview was conducted for approximately 20 min, and recorded for analysis after agreement was received from interview participants. Within the content of the consent form, it was specified that photos would be taken for research purposes. To prevent any problems regarding privacy protection, we made corrections by masking their faces.

When describing the heuristics evaluation, more details would be important. What heuristics exactly were used. The paper shows only a list of 22 keywords, but these are not understandable as heuristics.

=> As you have pointed out, there were parts of the manuscript that were difficult to understand because the entire content was presented in a too compact manner. This aspect has been revised with supplementation of the content by the heuristics evaluation.

Where did the evaluators conduct the analysis? They they go up to the observatory? How long did it take to make the analysis?

=> This aspect has been revised with additional description in Page 24.

Round 2

Reviewer 1 Report

Fix the below concerns cautiously,

-I still believe there are many unnecessary  small paragraphs. 
Reduce the no. of paragraphs by merging few paragraphs into one
as deemfit.

-I still believe through proofread has not been done cautiously.

-The subsection name,`Research range and methods,' is insignificant and not standard; replace by `Major contributions'.

- Below comment has not been addressed at all, although it renamed in the revised vesrion as Figure 11.

Figure 4 consits of 4 sub-figures. Hence at first separate them considering  4(a), 4(b) etc.

Use the figure caption at the bottom of each figure, rather than on top. undefined Y-axis!

-The sebsection `Research background and goal' needs substantial improvement.

-Presentation of the paper needs improvement.

- used references are not standard. 

-inadequate literature survey

-OMG, What is this `AR((Augment Reality),'??????

-Results are not properly demonstrated.

-needs more results as like Figure 11, otherwise it looks like a survey paper.

-enourmous possibility to improve the paper.

-NOvelty is not clear to me. very very important concern. otherwise prepare a survey paper.

***((Relect!!! My most of the comments have not been addressed and also not 
mentioned the reason in the Response letter.

Author Response

Thank you for reviewing our paper in detail.

-I still believe there are many unnecessary small paragraphs. Reduce the no. of paragraphs by merging few paragraphs into one as deemfit.

In our last revision, there was some confusion regarding your suggestion, and we focused on reducing the number of subtitles, rather than combining small paragraphs. As suggested, we have tried to merge related and connectable paragraphs.

-I still believe through proofread has not been done cautiously.

The paper was proofread by a native English speaker; however, it seems that it was not sufficient in the first round. Consequently, we have used MDPI’s English editing service.

-The subsection name,`Research range and methods,' is insignificant and not standard; replace by `Major contributions'.

=> The suggested correction has been implemented

-- Below comment has not been addressed at all, although it renamed in the revised version as Figure 11.

Figure 4 consists of 4 sub-figures. Hence at first separate them considering 4(a), 4(b) etc.

In the first review, a reviewer mentioned that the application result screens in the four sub-figures in Figure 4 were not clearly visible. In our first revision, we divided the images into Figures 4 to 9 including one intermediate process image, which had been explained only as part of the text before, in addition to the existing four sub-figures in Figure 4. The images were resized and a step-by-step explanation was given. For this reason, the figure numbers were changed. We have reflected this in our last revision, but it seems that there was some confusion.

Use the figure caption at the bottom of each figure, rather than on top. undefined Y-axis!

In our first revision, we defined X- and Y-axis only in the caption of Figure 11. This must have made it difficult to see the changes. In this revision, we have revised the X-axis and Y-axis details in the main text as well. Regarding the location of figure captions, we realized that the layout was incorrect after submitting our first revised version, and then we submitted another revised version to the journal editor. Perhaps, you checked the wrong layout in our first revised version. We have placed the figure captions below the figures and table captions above the tables in accordance with the formatting guidelines.

-The sebsection `Research background and goal' needs substantial improvement.

The contents of this section have been revised to reduce unclear expressions and provide more detailed explanation, in the first revision. Additional improvements have also been made in the second revision, where contents regarding novelty, focusing on major contributions, have been added.

-Presentation of the paper needs improvement.

In the first revision, we focused on rearranging figures and tables so that they are more legible. We assumed that the presence of small paragraphs is also an issue with presentation. Hence, we tried to fix this problem.

- used references are not standard. 

We have corrected the references in accordance with the formatting requirements of the journal.

-inadequate literature survey

This study is regarding an app for obtaining information about tourist attractions using AR on smart devices such as mobile phones. Thus, with respect to existing works, we examined studies on AR tour mobile applications and their characteristics. To design the observatory contents, we investigated the contents of three representative observatories in Seoul, the U.S., and Japan. Because it was difficult to find studies related to contents being provided in observatories, we collected data by visiting the observatories and from news articles regarding the observatories in South Korea. For the observatories located in other countries, we collected information from their websites and news articles. We were able to find cases related to AR mobile tour applications in some of the papers, but they were more business-related than academic. Thus, we collected information about the apps from news articles and product websites.

We could not find examples of AR tours using the views in observatory spaces in high-rise buildings from the data we collected. Most AR tours provided location-based services (LBS) in the actual tourist attractions.

-OMG, What is this `AR((Augment Reality),'??????

Seoul AR is an app that provides users with tourist information about famous tourist attractions in Seoul including their directions and locations in the observatory space using 50 landmark icons in the Seoul Sky Observatory as AR markers. The AR app provides a simple game in which landmark icons can be recognized through the cameras embedded in mobile devices. When they are successfully recognized, users can find the actual tourist attractions from the landmark icons through star-shaped AR objects. This app helps users to view the tourist attractions from the observatory and provides tourist the related information. Compared with the various AR features currently in the market, this app has a simple form that uses images as markers. However, we tried to provide information about various tourist attractions in Seoul easily and interestingly for visitors of all age groups while they enjoy a 360 degree-view of Seoul in the Seoul Sky Observatory, which is a representative landmark of Seoul.

-Results are not properly demonstrated. -needs more results as like Figure 11, otherwise it looks like a survey paper.

We have further elaborated on the results.

-enourmous possibility to improve the paper.

A significant effort has been made to improve the paper based on the provided comments.

-Novelty is not clear to me. very important concern. otherwise prepare a survey paper.

This study was conducted while the first author was working as a media exhibition manager at the Seoul Sky Observatory in 2019. At that time, the research was conducted while developing an AR mobile application for use in the Seoul Sky Observatory to increase visitor’s satisfaction level.

This paper is intended to be submitted to the Special Issue "LifeXR: Concepts, Technology and Design for Everyday XR" of Electronics, which is accepting submissions over a wide range of topics, such as user experience of mobile XR/ LifeXR, LifeXR applications, LifeXR in everyday spaces, and design solutions to LifeXR. Therefore, the authors consider the paper to be relevant to topic.

The study investigated interaction design, which is different from the scope of Electronics; therefore, we weighed on the possibility and relevance of the submission of this paper to the Special Issue "LifeXR: Concepts, Technology and Design for Everyday XR" of Electronics. After discussions with the journal editor, it was decided that the paper was pertinent to the themes covered in the Special Issue; thus, we have submitted the paper accordingly.

From a technical perspective, the contribution may not be novel because we have used Vuforia and Unity in this study, which are existing platforms commonly employed for AR mobile application development. However, in our opinion, the process of designing an AR application for use in the observatory space in a skyscraper is rare compared with the types of spaces supporting various AR services. The issues that have been identified through trial and error in the design process are thought to be helpful not only to the authors’ research team but also to other researchers who plan to develop AR applications for observatory spaces in the future.

As reviewed in this study, different observatory types are in operation in high-rise buildings in several countries, such as the United States and Japan. They differ in terms of operations and the ways in which they accommodate visitors and their activities. Given that there is no existing case of using an AR application for operations in skyscrapers, we believe this study will serve as an interesting example of AR application design in the Seoul Sky Observatory for researchers who wish to explore AR applications in this area. It may be difficult to generalize the design process and method employed in this study when developing AR applications for different types of observatories in various skyscrapers. However, both researchers and readers may find that the study is helpful in terms of designing a mobile AR application (referred to as Seoul AR). The application was developed to enhance visitor satisfaction at the Seoul Sky observatory.

***((Relect!!! My most of the comments have not been addressed and also not mentioned the reason in the Response letter.

The majority of the suggestions have been addressed in the first revision; however, a more concerted effort has been made to clearly show the revisions made.

Reviewer 3 Report

I appreciate the effort that the authors put into the revised version in order to try to address most comments. The revised version is clearer and more complete.

However, my main objection is the lack of a clear scientific contribution. I see no novelty in approach, technical solution, methodology, etc., that might justify publication in a scientific journal. In fact, the proposed app does not even seem to satisfactorily solve the issue of providing information about the landmarks given the rough approximation that was made to an average user height!

In addition, I also have doubts that the need of the application follows from the initial user research. Users complained about the difficulty of getting information about the landmarks of identifying them and the app seems to make it hard also to get this information by forcing users to play a game. The main purpose seems not to provide a service to the user by facilitating information access but to make them spend more time at the observation tower.

Author Response

However, my main objection is the lack of a clear scientific contribution. I see no novelty in approach, technical solution, methodology, etc., that might justify publication in a scientific journal. In fact, the proposed app does not even seem to satisfactorily solve the issue of providing information about the landmarks given the rough approximation that was made to an average user height!

Thank you for your careful feedback. As mentioned in the last rebuttal, the intention was to submit this paper to Electronics’ special issue "LifeXR: Concepts, Technology and Design for Everyday XR". Before submitting the paper, we checked with the journal editor regarding the suitability of our paper, which is centered on interaction design content, for the special issue and received a positive response.

The concern that the scientific contribution, technical solution, and methodology of our study are different from those published in Electronics is valid. However, as far as we know, the special issue is accepting papers on XR-related designs and applications as well as papers with technical contributions. Given that the theme is Life XR, we thought that various services and designs used in every space and situation of our lives are also topics that are covered.

https://www.mdpi.com/journal/electronics/special_issues/lifeXR_tech_design

Our search effort may have been insufficient, but we could not find examples of AR tour mobile apps that use views in observatories in high-rise buildings. Most AR tour mobile apps were based on LBS (Location Based Service), and we could not find examples of apps that connected the views of tourist attractions from an observatory. Our technical distinctions may have many shortcomings from an engineering viewpoint. However, our effort to apply AR to the service space where AR has not been applied until now could provide a reference to researchers who want to combine AR contents with views in observatories as well as for follow-up studies of our research team in the future.

We think that the reviewer’s understanding is different from our intention. After analyzing users’ needs for the Seoul Sky Observatory, we planned our app based on the assumption that contents are required for obtaining information needed for tourism in Seoul and experience in the observatory spaces. The developed app efficiently finds the directions and locations of tourist attractions in Seoul and receives related tourist information in the Seoul Sky Observatory, as we expected.

The problem regarding the actual directions of landmark areas are due to the differences in user height, smartphone position, and the distance between the user and AR object, as explained in “Design revision for the final prototype,” was discovered through expert opinion in the Heuristic evaluation process. To reduce potential problems due to errors in actual user experiments, the locations of actual AR objects were corrected in the final experiment by finding the average male/female heights and face heights of Korean subjects in their 20s to 50s. We wanted to highlight this.

It is not that the designed app did not work or was different from our initial expectations. During the experiment, we discovered that the deviation caused by the long distance could increase due to the nature of the observatory. We corrected this deviation as much as possible and applied it to the user experiment. As mentioned in the Discussion, this issue provided us with an opportunity to identify a method to locate real places from the observatory by using AR and image processing together, rather than using fixed AR markers such as landmark icons in the observatory in follow-up studies.

This study addressed an issue that we discovered while conducting the research. It was not conducted to solve a problem that we discovered. As you mentioned, the lead author, who worked as a member of the media exhibition staff in the Seoul Sky Observatory, conducted this study to improve the satisfaction of observatory visitors and increase their stay in the observatory by providing users with observatory experience contents as well as information about major tourist attractions in Seoul, while they enjoy the views of Seoul. 

In addition, I also have doubts that the need of the application follows from the initial user research. Users complained about the difficulty of getting information about the landmarks of identifying them and the app seems to make it hard also to get this information by forcing users to play a game. The main purpose seems not to provide a service to the user by facilitating information access but to make them spend more time at the observation tower.

in Seoul while enjoying the complete view of Seoul as well as the contents of the observatory; however, the problem was that the observatory currently does not provide introductions or related information about tourist attractions in Seoul. The observatory services were provided in a high-rise building of a private company, not in a public building in Seoul. As such, they did not have to provide information about tourist attractions. However, the visitors of the observatory wanted to see tourist attractions in Seoul from the observatory, where they could see the entire city, or to find related information because it is a representative landmark.

As can be seen from the user interviews, even though there was no limitation to the length of stay in the observatory, most users just descended after experiencing most of the facilities in the observatory for which they paid an expensive entrance fee and waited for a long time. A problem was that the contents that visitors can enjoy were few. Expecting to increase users’ length of stay in the observatory and thereby improve satisfaction, the observatory wanted to provide information about the tourist attractions that users want to find, using 50 landmark icons for major tourist attractions that can be seen from the observatory.

To attract users’ attention and induce fun, simple game elements were introduced with the intention being not to force games or give discomfort to users. For example, as mentioned in the system description, in Mission 2, if a user makes an inaccurate selection of the corresponding place, hints are provided to allow the users to easily find the corresponding place. In addition, if the user successfully finds the place by completing one mission, all landmark icons that were not cleared in the game are activated to allow users to find information about the tourist attractions that they want to check. Thus, users can directly click the activated landmark icons for the places they want to know about any time without playing additional games. The game elements have been added not to make users uncomfortable or force them to play games, but to induce users’ interest and curiosity.  

Round 3

Reviewer 1 Report

Below few minute concerns:
-AR(Augment Reality) ---> Augment Reality(AR)
-I still believe through proofread has not been done cautiously.
-Include few relevant recent references from MDPI journals.

Author Response

AR(Augment Reality) ---> Augment Reality(AR)

  • We changed AR (Augment Reality) to Augment Reality (AR) in 3rd revision. 

I still believe through proofread has not been done cautiously.

  • Our 3rd revision received English correction and proofreading services through MDPI. After receiving the correction service, the overall manuscript has been improved. Thank you so much.

Include few relevant recent references from MDPI journals.

  • Recently published researches, related to AR mobile and AR tourism, in the MDPI journals, were supplemented in the related research section. We can see the content improved after that. Thank you so much for your advice.

Reviewer 3 Report

I believe that both the authors position and opinions and mine are clear now and that the editor can weigh them in the final decision.

I maintain my position expressed in the previous review, despite the arguments offered by the authors.

Author Response

I believe that both the authors position and opinions and mine are clear now and that the editor can weigh them in the final decision. I maintain my position expressed in the previous review, despite the arguments offered by the authors.

We want to appreciate that you have worked hard to review our manuscript in the meantime. We also fully understand and respect your opinions, but we also believe this manuscript will enrich the Special Issue "LifeXR: Concepts, Technology, and Design for Everyday XR". Thank you for the generous help in the improvement of this manuscript.
